



# Validation of digital elevation models (DEMs) and comparison of geomorphic metrics on the southern Central Andean Plateau

Benjamin Purinton[1] and Bodo Bookhagen[1]

1, Institute of Earth and Environmental Science, Universität Potsdam, Potsdam, Germany

*Correspondence to*: Benjamin Purinton (purinton@uni-potsdam.de)

**Abstract.** We validate and compare elevation accuracy and geomorphic metrics of the current generation of satellite-derived digital elevation models (DEMs) on the southern Central Andean Plateau. The plateau has an average elevation of 3.7 km, and is characterized by diverse topography and relief, lack of vegetation, and clear skies that create ideal conditions for remote sensing. At 30 m resolution, the SRTM-C, ASTER GDEM2, stacked ASTER L1A stereopair DEM, ALOS World 3D, and

TanDEM-X have been analyzed. The higher resolution datasets include 12 m TanDEM-X, 10 m single-CoSSC TerraSAR-X / TanDEM-X DEMs, and 5 m ALOS World 3D. We assessed vertical accuracy by comparing standard deviations (SD) of the DEM elevation versus 307,509 differential GPS (dGPS) measurements across 4,000 m of elevation. Vertical SD for the 30 m DEMs were 9.48 m (ASTER GDEM2), 6.93 m (ASTER Stack), 3.33 m (SRTM-C), 2.81 m (ALOS World 3D), and 2.42 m (TanDEM-X). Values were generally lower for higher resolution DEMs at 2.02-3.83 m (10 m single-CoSSC TerraSAR-X /

TanDEM-X), 1.97 m (12 m TanDEM-X), and 1.64 m (5 m ALOS World3D). Analysis of vertical uncertainty with respect to terrain elevation, slope, and aspect revealed the high performance across these attributes of the 30 m SRTM-C, 30 m and 12 m TanDEM-X, and 30 m and 5 m ALOS World 3D DEMs. Single-CoSSC TerraSAR-X / TanDEM-X 10 m DEMs and the 30 m ASTER GDEM2 displayed slight aspect biases, which were removed in their stacked counterparts (TanDEM-X and ASTER Stack). Based on high vertical accuracy and visual inspection of minimal hillslope artifacts alongside optical satellite data, we

selected the 30 m SRTM-C, 12-30 m TanDEM-X, 10 m single-CoSSC TerraSAR-X / TanDEM-X, and 5 m ALOS World 3D for geomorphic metric comparison in a 66 km² catchment with a distinct river knickpoint. For trunk channel profiles analyzed with chi plots, consistent *m/n* values of 0.49-0.57 were found regardless of DEM resolution or SD. Hillslopes were compared through slope and curvature calculations to assess basin-wide differences in their distributions and in the hillslope-to-valley transition related to the knickpoint feature. We find 0.1-0.2 m/m higher slopes downstream of the knickpoint, related to the

over-steepened baselevel. While slope and hillslope length measurements vary little between datasets, curvature displays higher magnitude measurements with fining resolution. This is especially true for the optical 5 m ALOS World 3D DEM, which demonstrated high-frequency noise in 2-8 pixel steps through a Fourier frequency analysis. The improvements in accurate space-radar DEMs (e.g., TanDEM-X) for geomorphometry are promising, but airborne or terrestrial data is still necessary for meter-scale analysis.



# 1. Introduction

Digital elevation models (DEMs) provide hydrologists and geomorphologists with powerful tools to explore the linkages between fundamental geomorphic processes and landforms and to test hypotheses of landscape evolution at local and regional scales using geomorphic metrics (e.g., Howard et al., 1994; Burbank et al., 1996; Whipple and Tucker, 1999; Montgomery et al., 2001; Dietrich et al., 2003; Roering et al., 2007). Modern geomorphologists use the quantitative sub-discipline of geomorphometry (Pike et al., 2009; Wilson, 2012) to explore how tectonic, climatic, and lithologic signals can be inferred from DEMs (e.g., Snyder et al., 2000; Wobus et al., 2006; DiBiase et al., 2010; Bookhagen and Strecker, 2012; Kirby and Whipple, 2012; Scherler et al., 2015; Clubb et al., 2016; Olen et al., 2016), but questions remain to what extent transient responses can be recorded in landscape morphology (e.g., DiBiase et al., 2012) and how channel networks and hillslopes can independently act as records of basin transience (e.g., Ouimet et al., 2009; Hurst et al., 2012; Clubb et al., 2016; Forte et al., 2016). Such studies rely on accurate DEMs for the calculation of geomorphic metrics (e.g., slope and curvature) and extraction of geomorphic features (e.g., channels, hillslopes, hilltops). In spite of this, DEM elevation error reporting (Fisher and Tate, 2006; Reuter et al., 2009) – often carried out with limited control data – only accounts for absolute pixel elevation accuracy and does not include higher-order DEM derivatives (e.g., slope and curvature), geomorphic metrics, or landscape features of interest to geomorphologists. This problem is especially acute given that relatively small elevation errors will propagate and grow in the first (slope) and second (curvature) derivatives, potentially obscuring geomorphometric results (e.g., Wechsler, 2007).

Remotely sensed DEMs – referred to throughout this study as DEMs, as opposed to the often used term digital terrain model (DTM) for bare-earth models with vegetation and structures removed – are generated from data that are originally distorted through sensor, terrain, and atmospheric conditions leading to misrepresentations (error) in the final product (Smith and Sandwell, 2003; Fisher and Tate, 2006; Nuth and Kääb, 2011). Additionally, these datasets are typically received in gridded raster format – rather than point cloud or triangulated irregular network (TIN) – resulting in a defined measurement interval (pixel size) that may oversimplify fine landscape variability. Thus, the geomorphic scales of interest must be taken into consideration when selecting the appropriate DEM (e.g., Hengl, 2006). For instance, while channel profiles over long reaches are readily analyzed on 90 m resolution data, hillslopes with considerably smaller extents require higher 1-30 m resolution data capable of identifying individual hillslopes and ridges (Grieve et al., 2016a,b,c). Furthermore, DEM biases specific to a given sensor should be considered prior to analysis, especially when using satellite-derived DEMs in steep topography (e.g., Paul and Haeberli, 2008; Nuth and Kääb, 2011).

Since the release of the first global DEM by the United States Geological Survey (USGS) in 1996 (GTOPO30) at a resolution of 30 arc-seconds (~1 km), advances in remote sensing technology – particularly satellite observation – and processing capabilities have steadily improved the accuracy and reduced the resolution of DEMs. The 2003 release of the 90 m Shuttle Radar Topography Mission (SRTM) DEM with coverage from 56° S to 60° N (Farr et al., 2007) ushered in a new age of near-global digital topographic analysis (Wilson, 2012). With the 2009 release of the 30 m Advanced Spaceborne Thermal Emission



and Reflection Radiometer Global DEM (ASTER GDEM; ASTER, 2009), and more recent releases of the improved ASTER GDEM version 2 (ASTER GDEM2; Tachikawa et al., 2011), SRTM C-band 30 m (SRTM-C), and up-sampled Advanced Land Observing Satellite (ALOS) World 3D 30 m (AW3D30), geomorphologists now have open-access to many 30 m near-global DEMs. In addition to these public 30 m datasets, higher resolution (< 15 m) DEMs from a variety of satellite sources

are becoming increasingly available through commercial purchase or research agreements as edited products (e.g., ALOS World 3D and TanDEM-X WorldDEM), optical pairs for stereogrammetric processing (e.g., ALOS Panchromatic Remote-sensing Instrument for Stereo Mapping, or PRISM), and radar scenes for interferometric processing (e.g., TerraSAR-X / TanDEM-X).

The SRTM-C and ASTER GDEM2 have reported vertical accuracies of ~5-20 m depending on terrain characteristics (e.g.,

Mukherjee et al., 2013; Rexer and Hirt, 2014), with some biases reported related to slope and aspect of the terrain (e.g., Berthier et al., 2006; Nuth and Kääb et al., 2011; Shortridge and Messina, 2011). While these accuracies allow long-term (decadal) tracking of glacial elevation changes (e.g., Racoviteanu et al., 2007; Paul and Haeberli, 2008), higher resolution local DEMs from optical and radar sources have proven more accurate (< 5 m vertical error) than these global products for glacial studies in steep terrain, particularly on shorter time scales (e.g., Berthier et al., 2007; Berthier and Toutin, 2008; Jaber et al., 2013;

Neckel et al., 2013; Pandey and Venkataraman, 2013; Holzer et al., 2015; Rankl and Braun, 2016; Neelmeijer et al., in review). However, to date no studies have assessed the accuracy of the current generation of sub-15 m, satellite-derived DEMs with regards to geomorphometry. These measurements, unlike glacial studies, rely on the derivatives of elevation (e.g., slope and curvature) and their spatial context, not absolute or relative height changes. Furthermore, glacial studies are typically conducted on lower slope terrain and compare area-wide measurements allowing some error to average out. On the other hand,

geomorphic studies examining channels and hillslopes in steeper terrain may be more impacted by remote-sensing errors and artifacts (e.g., from shadowing, sensor angle, foreshortening), and geomorphic metrics like slope and curvature rely on the accuracy of nearby pixels, for example within a 3×3 moving window.

The application of light detection and ranging (lidar) by ground and aerial methods is often used to generate meter to sub-meter scale elevation point clouds and gridded DEM datasets at smaller areal extents than satellite-derived DEMs (e.g.,

Passalacqua et al., 2015). Lidar has revolutionized geomorphology with more accurate representations of the land surface and led to new insights and discoveries in the realm of mass and energy transport laws, channel initiation, surface flow routing, and landslide and fault scarp mapping (e.g., Dietrich et al., 2003; Roering et al., 2007; Roering et al., 2013; Perroy et al., 2010; Shelef and Hilley, 2013; Tarolli, 2014). While coarser DEMs have proven useful in exploring mountain belt hypsometry and linkages between climate, erosion, and tectonics at basin or regional scales (e.g., Montgomery et al., 2001; DiBiase et al., 2010;

Bookhagen and Strecker, 2012), their utility in analyzing process-level geomorphology and assessing critical hillslope parameters is limited and lidar is often deemed necessary (Perroy et al., 2010; Roering et al., 2013; Tarolli, 2014; Passalacqua et al., 2015). Despite this, the limited spatial extent (~1 km$^2$) and high effort and cost of obtaining lidar are prohibitive factors to its application at basin or regional scales (10-1,000 km$^2$).



Previous studies examining the effect of DEM resolution on geomorphic metrics and features have primarily used resampled or re-gridded lidar data (e.g., Tarolli and Tarboton, 2006; Tarolli and Dalla Fontana, 2009; Grieve et al., 2016c). Here we are interested not in high-quality resampled data, but rather data at their original resolution collected from different sensors, without any higher-resolution information from resampling. Advances in sub-15 m DEM availability and accuracy from a

number of satellites necessitates investigation of their advantages over 30 m public DEMs in representing derivatives of elevation for channel and hillslope analysis in lieu of lidar.

This study presents a multi-DEM validation and comparison for the southern Central Andes in NW Argentina in an arid landscape with no vegetation cover, ideal for remote sensing. DEM validation is presented by: (i) reporting the vertical accuracy of a number of satellite-derived DEMs at resolutions of 5-30 m from open-access portals, commercial sources, and

research agreements; and (ii) carrying out channel profile analysis and geomorphic metric comparisons for a 66 km$^2$ catchment with a defined channel knickpoint to assess the quality of these DEMs for tectonic geomorphology. Through this analysis we demonstrate the state-of-the-art in wide-coverage, satellite-derived DEM availability for geomorphometry.

## 2. Study Area

The Puna de Atacama plateau in NW Argentina (Fig. 1A) is the southern extension of the low relief, high elevation, internally

drained Central Andean Plateau (also referred to as the Altiplano-Puna Plateau), extending for over 1,500 km and reaching widths of over 350 km in the Central Andes (Allmendinger et al., 1997). Due to the plateau's hyper-arid climate caused by orographic blocking and regional atmospheric circulation patterns (Bookhagen and Strecker, 2008; Rohrmann et al., 2014), there is an absence of cloud and vegetation cover on the Puna, creating ideal conditions for remote sensing of the bare-earth surface. As the Puna is largely uninhabited and erosion rates are very low (Bookhagen and Strecker, 2012), the study site is a

pristine environment experiencing little change from year-to-year, thus minimizing differences between DEMs collected years apart. Topographic expression is diverse on the plateau with flat salars (salt flats) having near-zero relief at 5-10 km scales surrounded by steep volcanoes and mountain ranges with > 2 km of relief at 2-5 km scales. This morphology is readily apparent around the Pocitos Basin, centered on the Salar de Pocitos (basin elevation at ~3,600 m) and bordered by mountains such as the Nevado Queva reaching elevations of over 6,000 m (Fig. 1B). Within the Pocitos Basin, we focus geomorphometric

analysis on the 66 km$^2$ Quebrada Honda catchment, with 1.2 km of relief (Fig. 1C). The Quebrada Honda was chosen for its size, coverage across available DEMs, uniform Paleozoic metasedimentary lithology, and the presence of a knickpoint 7 km upstream of the outlet dividing the basin into transiently adjusting sections.







**Figure 1. (A) Topographic overview of the study area in the southern Central Andes. 307,509 dGPS measurements displayed in pink. UNSA base station (white star) for dGPS kinematic correction located in Salta, Argentina. Inset shows South American continent with international borders and internally-drained Central Andean Plateau. Study focus is the Pocitos Basin (B), where elevation ranges from 3,600 m on the flat salar to 6,000 m on surrounding peaks. Geomorphometric analyses focus on the Quebrada Honda (C) catchment draining an area of 66 km² from 5,000 m of elevation down to 3,800 m. A knickpoint 7 km upstream divides the basin into an upper and lower section with differing morphology (Fig. 2). The transition is observable along the trunk as normalized channel steepness ($k_{sn}$) averaged along 300 m reaches on the SRTM-C 30 m DEM increases to values > 500. The $m/n$ reference value of 0.52 is calculated using chi plot analysis. Elevations in (A) and (B) are from the 90 m SRTMv4.1 DEM (Jarvis et al., 2008).**

## 3. Data and Methods

### 3.1. dGPS Data

Vertical accuracy of optical and radar DEMs was assessed using a differential GPS (dGPS) dataset spanning ~4,000 m of elevation and covering an area of ~50,000 km² centered on the Pocitos Basin (Fig. 1A). Of 333,555 total raw dGPS



measurements collected during field campaigns from 2013-2016, 307,509 kinematically corrected points with vertical and horizontal accuracies < 0.5 m were selected for the final control on DEM vertical accuracy. Data were projected to the EGM96 vertical and WGS84 horizontal datums in the UTM coordinate system zone 19S. This point measurement dataset was rasterized to the resolution and extent of each DEM. Multiple measurements within a DEM pixel were averaged and pixels without

measurements were set to no data. This led to a reduction in the number of individual measurements used to assess DEM vertical accuracy, but accounted for multiple measurements per pixel to provide a robust validation. Details of measurement collection and kinematic correction of the raw dGPS files using the UNSA permanent station in Salta (Fig. 1A) can be found in the Supplement.

## 3.2. DEM Datasets

DEMs collected from a number of public, commercial, and research agreement sources are listed in Table 1. All were referenced to the same datums (EGM96 / WGS84) and projected into UTM 19S using bilinear interpolation. DEMs were co-registered to a common control – the 30 m SRTM-C, selected for its excellent geolocation accuracy (Rodriguez et al., 2006) – using affine parameters by up- or down-sampling the SRTM-C to the resolution of the DEM of interest and iteratively shifting to reduce the root mean squared error (RMSE) of the elevation difference using the Matlab function *imregister*. By aligning

datasets to one another, we minimize elevation uncertainty versus the dGPS measurements caused by slight offsets in geolocation of the DEMs (e.g., Nuth and Kääb, 2011). Co-registration was not carried out on the TanDEM-X data, as the SRTM-C was used during initial processing steps to provide elevation corrections (Wessel, 2016). Additional information on each dataset listed in Table 1 are found in the Supplement, including datasets that were not included in the rest of the study due to lower resolution, lack of coverage, or quality issues (SRTMv4.1 90 m, SRTM-X 30 m, RapidEye 12 m, SPOT6 5 m,

ALOS PRISM tri-stereopair 10 m, TerraSAR-X pairs 10 m, and single-CoSSC TerraSAR-X / TanDEM-X processed to 5 m).



**Table 1. List of DEMs used for comparisons and geomorphic analyses.**

| Dataset (short name) | Data Type | Resolution (m) | Source | Notes |
|---|---|---|---|---|
| SRTM C-band (SRTM-C) | Radar / Edited global product | 30 | Public / https://lta.cr.usgs.gov/SRTM1Arc | Released 2014, previously only US coverage. |
| ASTER GDEM Version 2 (ASTER GDEM2) | Optical / Edited global product | 30 | Public / https://asterweb.jpl.nasa.gov/gdem.asp | Released 2011, update of ASTER GDEM1 released 2009. Generated by automated processing and stacking of ASTER L1A stereopairs by NASA and METI.[a] |
| ASTER L1A Stereopair Stack (ASTER Stack) | Optical / Raw stereopairs | 30 | Public / http://reverb.echo.nasa.gov/reverb/ | Stacked DEM generated here by manual stereogrammetric processing of eight raw L1A stereopairs (Sect. 3.2.1.). |
| ALOS World 3D (AW3D5 and AW3D30) | Optical / Edited global product | 5/30 | Public (30 m) / http://www.eorc.jaxa.jp/ALOS/en/aw3d30/  Commercial (5 m) / http://aw3d.jp/en/ | 5 m DEM released 2015 as highest resolution commercial global DEM with down-sampled 30 m research version released 2016. |
| Single-CoSSC [b] TerraSAR-X / TanDEM-X (CoSSC TDX)[a] | Radar / Raw interferograms | 10 | Research agreement / http://terrasar-x.dlr.de/ | CoSSC TerraSAR-X / TanDEM-X mission radar pair DEMs were used by DLR to generate the stacked TanDEM-X DEM in 2015. |
| TanDEM-X DEM (TanDEM-X 12 m and TanDEM-X 30 m) | Radar / Edited global product | 12/30 | Research agreement / http://tandemx-science.dlr.de/ | Final 12 m DEM generated by stacking of overlapping CoSSC TerraSAR-X / TanDEM-X radar pair DEMs and down-sampled 30 m version, both from DLR.[c] |

a, Ministry of Economy, Trade and Industry (METI) of Japan
b, Co-registered Single Look Slant Range Complex (CoSSC) raw interferometric product from DLR
c, Commercial 12 m product available as WorldDEM™ from AIRBUS

### 3.2.1. ASTER Stacking

The ASTER radiometer has collected along-track stereopairs with nadir (Band 3N) and backward (Band 3B) looking near

5   infrared cameras between 83º S and 83º N since 1999 (Tachikawa et al., 2011). Using these stereopairs, a 30 m ASTER global

DEM has been generated by automatic stereo-correlation, stacking, and averaging of over 1.2 million scenes. The stacking of





multiple lower quality DEMs from the same source is a common technique, also undertaken to generate the 12 m TanDEM-X (from single-CoSSC TDX DEMs) and 5 m ALOS World 3D (from ALOS PRISM optical tri-stereopair DEMs). The 2011 release of the ASTER GDEM version 2 (ASTER GDEM2) used in the present study represented a vast improvement in quality (Tachikawa et al., 2011), but remaining noise is caused by issues with cloud cover, water masking, the smaller stereo
correlation kernel, and mis-registration of scenes prior to stacking (Nuth and Kääb, 2011). We seek to improve on the ASTER GDEM2 using eight raw ASTER L1A 3N/B stereopairs downloaded with variable overlap from the Pocitos Basin. Using stereogrammetric processing methods we generated eight 30 m DEMs from these stereopairs. Details of DEM generation along with RMSE of ground control and tie points are presented in the Supplement (Table S1). Each L1A DEM was co-registered to the SRTM-C, manually masked for outliers (locations where clouds or haze in the imagery caused abrupt > 1,000 m steps
in the final DEM), and differenced with the SRTM-C. Pixels were weighted with a bi-square scheme based on their correlation with the SRTM-C, and a weighted average of the overlapping DEMs was used to generate a higher quality 30 m ASTER Stack.

### 3.3. Elevation Accuracy Assessment

To assess DEM vertical accuracy, we first performed a pixel-by-pixel comparison of rasterized dGPS (vertical uncertainty < 0.5 m) and DEM elevation values after co-registration to the SRTM-C. Our preferred metric for DEM vertical accuracy is the
mean ± 1-sigma ($\sigma$) standard deviation (SD) (Li, 1988; Fisher and Tate, 2006). Specifically, we are interested in the SD of DEM elevation versus dGPS height as our quality metric. Plotted histograms of uncertainty distribution were normalized by their respective mean offsets so the SD could be visually compared. Differences of ±30 m were filtered out as outliers caused by bad data and processing errors, and the percentage reduction in number of measurements from this filtering is reported as an additional quality check. In a second step, we examined error distributions with respect to terrain slope, aspect, and elevation
for the DEMs, also normalized by mean offset with ±30 m outliers excluded. Measurements were separated into 50-100 m elevation bins (depending on the full elevation range of the dataset), slopes were calculated by their eight-connected neighborhood and binned by 1°, and aspect (also eight-connected calculated) was binned by 10° with north at 0° and east at 90°. Vertical uncertainty was plotted in each bin as a box plot showing the median, 25-75th percentile range, and 1st and 99th percentile outlier cutoffs.

### 3.4. Geomorphometric Analysis

For a robust assessment of DEM quality, we go beyond pixel-by-pixel vertical accuracy comparisons by comparing longitudinal channel profiles and derived geomorphic metrics in the 66 km$^2$ Quebrada Honda catchment (Fig. 2). Here, a defined channel knickpoint separates downstream steep and upstream gentle-sloped terrain, and consistent climate and lithology allows us to test hypotheses of basin-wide adjustment to river gradients. We focused on a subset of the highest quality
DEMs with the aim to provide an assessment of the effects of different sensors and resolutions (e.g., SRTM-C 30 m, TanDEM-X 12 m and 30 m, and AW3D5 5 m) and DEM stacking (e.g., CoSSC TDX 10 m and TanDEM-X 12 m) on geomorphometry.







**Figure 2. Topographic overview of Quebrada Honda from 12 m TanDEM-X DEM. (A) Normalized channel steepness ($k_{sn}$) averaged over 300 m reaches using $m/n = 0.52$ with upstream and downstream drainage areas indicated by black outlines. All tributaries with drainage area > 1 km² are plotted. Note over-steepened trunk signal has not propagated entirely up all downstream tributaries, as indicated by $k_{sn} < 400$ in upper reaches. (B) Longitudinal profile of trunk channel and tributaries with knickpoint indicated. (C) D∞ slope map (Tarboton, 2005) displaying steeper, more variable topography downstream of knickpoint, indicated by warmer colors and greater average slope ($S_{ave}$) and standard deviation. (D) Curvature colored by ±3-σ range with positive values concave (valleys) and negative values convex (hilltops). Note the planar slopes separating ridges and valleys and the increase in concavity near valley heads, indicating the shift from hillslope to fluvial processes.**

### 3.4.1. Channel Profile Analysis

Hydrological and geomorphic modeling is an important application of DEMs (Wilson, 2012) and channel network extraction is a necessary step prior to channel profile analysis. Although a number of recently developed methods for channel extraction via channel head identification now exist (see Hooshyar et al. (2016) for a review), these methods have all been developed on high resolution lidar data, with control datasets of field-mapped channel heads (Clubb et al., 2014) or channel networks





(Passalacqua et al., 2010a,b). We are thus wary to apply these methods to our coarser satellite-derived data with no control from the field and no high resolution lidar data for a relative performance assessment. Instead we use the simplistic threshold area approach (e.g., Tarboton et al., 1991), choosing the common reference area of 1 km$^2$ where breaks in slope-area scaling indicate the changeover to dominantly alluvial channel processes (Montgomery and Foufoula-Georgiou, 1993).

Misrepresentation of channel location from this method is entirely restricted to the highest catchment reaches where the channel head lies, and does not affect the majority of the downstream channel. The consistent use of area thresholding at the same reference area across DEMs allows the direct comparison of channel profile results in our study.

Advances in longitudinal channel profile analysis driven by accurate DEMs have elucidated changes in boundary conditions recorded in channel slope and upstream propagating knickpoints (e.g., Wobus et al., 2006; Kirby and Whipple, 2012). The

stream power incision model (SPIM) of landscape evolution provides the theoretical basis for relating channel slope and drainage area (see Kirby and Whipple (2012) or Lague (2014) for background and limitations of SPIM). Applied to a channel profile in steady state ($dz/dt = 0$) we find the relationship:

$$S_C = \left(\frac{U}{K}\right)^{\frac{1}{n}} A^{\frac{-m}{n}} , \tag{1}$$

where $U$ is uplift, $K$ is erodibility, $A$ is local drainage area, $S_C$ is local channel slope, and $m$ and $n$ are site-specific constants

that scale the relative influences of climate and tectonics. Important to constrain here is the $m/n$ ratio, used to normalize channel steepness across differently sized drainage areas for the mapping of regional patterns of deformation, climatic influence, and/or lithologic boundary conditions (e.g., Wobus et al., 2006; Kirby and Whipple, 2012; Forte et al., 2016). While Eq. (1) is derived for steady state, its integration (or the use of area-slope plots) to determine the $m/n$ ratio and assess relative differences in channel steepness is a geometric consideration of local channel behavior and can thus be applied in non-equilibrium settings,

like the Quebrada Honda. Here, we utilize the recently developed integration method of chi plot analysis (Perron and Royden, 2013) to estimate $m/n$ from our DEMs (see Supplement for details).

Following channel extraction, we first applied the least-squares $R^2$ maximization chi plot technique of Perron and Royden (2013) to the Quebrada Honda trunk stream (Schwanghart and Scherler, 2014). This method attempts to linearize the entire channel to one best-fit line in chi space and does not provide robust uncertainty estimates for $m/n$, as linear regression is

performed through serially correlated values of chi distance and elevation (Perron and Royden, 2013). Because of this, we also employed the piece-wise fitting $m/n$ selection algorithm developed by Mudd et al. (2014) on the 30 m SRTM-C, 10 m CoSSC TDX, and 5 m AW3D5 DEMs (representing a cross section of DEM resolutions and sensors), for comparison with the least-squares approach. This method balances goodness-of-fit for the piece-wise fit profile with model complexity (number of parameters and segments) to provide an $m/n$ at the minimum corrected Akaike information criterion (*AICc*) (Akaike, 1974;

Hurvich and Tsai, 1989). A SD (uncertainty) of this minimum *AICc* is also provided, over which *AICc* values falling within the SD range indicate other plausible $m/n$ values (Mudd et al., 2014). Sensitivity tests were performed by varying fitting parameters with final parameters reported in the Supplement.





### 3.4.2. Hillslope Geomorphic Metrics

Besides channel profile analysis, signals of denudation and uplift may also be inferred from hillslope morphology as determined by geomorphic metrics including characteristic hillslope length, local relief, slope angles, and curvature. These parameters allow the exploration of empirical geomorphic transport laws, which aid in topographic modelling over geologic

timescales (cf. Dietrich et al., 2003). In particular, the accurate sampling of slope angles and curvatures allows patterns of erosion to be mapped from topography alone, thus playing key roles in geomorphic studies focused on the topographic expression of tectonic-climatic forcing (e.g., DiBiase et al., 2010; Hurst et al., 2012). Here, we test the newest generation of satellite-derived DEMs for assessing slope and curvature as well as the hillslope-to-valley transition marked by inflections in plots of slope, curvature, and drainage area to examine differences related not only to the channel knickpoint, but also to the

resolution and quality of the DEM. We compared results between the high quality 30 m SRTM-C, 30 m and 12 m TanDEM-X, 10 m CoSSC TDX, and 5 m AW3D5 DEMs. We did not include the ASTER DEMs in hillslope analyses because of elevation noise prevalent in these 30 m DEMs apparent in vertical uncertainty reporting (> 6 m) and visual inspection. Despite low vertical uncertainty (< 3 m), we also excluded the newly released 30 m AW3D30 because of unknown pre-processing steps taken to produce this DEM, which may have compounded high-frequency noise present in the 5 m AW3D5.

Since hillslopes represent a diffusive environment where flow is multi-directional, we calculated drainage area and slope at every pixel in the Quebrada Honda using the D∞ algorithm allowing dispersive flow (Tarboton, 2005). Curvature was calculated using the Laplacian of elevation (e.g., Tarolli and Dalla Fontana, 2009):

$$C = \nabla^2 z = \left(\frac{\delta^2 z}{\delta x^2} + \frac{\delta^2 z}{\delta y^2}\right), \qquad\qquad\qquad (2)$$

where concavity (valleys and channels) is denoted by $C > 0$, convexity (hillslopes and ridges) is denoted by $C < 0$, and planar

slopes are denoted by $C = 0$ (Fig. 2D). Distributions of slope and curvature separated upstream and downstream of the knickpoint were visualized as box plots displaying medians, 25-75[th] percentile ranges, 1[st] and 99[th] percentile cutoffs, and all outlier measurements.

Filtering is a common step to smooth DEM noise before deriving geomorphic metrics, often to reduce noise associated with shorter time-scale geomorphic features, like tree throw, from high-resolution lidar DEMs (Grieve et al., 2016a). During initial

tests we experimented with median, diffusion (Passalacqua et al., 2010b), and Wiener (Wiener, 1949) filtering prior to slope and curvature calculations. However, since all smoothing techniques were found to reduce the variability in slope and curvature measurements and blur sharper features of interest such as ridge-crests and valley bottoms, we instead chose to measure derivatives from the raw elevation data.

To explore the influence of the over-steepened trunk reach on hillslope morphology we combined measures of curvature

(Laplacian), slope (D∞), and drainage area (D∞) at every DEM pixel in the Quebrada Honda to explore differences between the gentle upstream and steep downstream catchment. We are particularly interested by differences in the hillslope-to-valley transition demarcated by the first inflection in plots of slope binned by area, occurring at a critical drainage area where channel heads are thought to initiate (Montgomery and Foufoula-Georgiou, 1993; Zhang and Montgomery, 1994; Ijjasz-Vasquez and



Bras, 1995; Tarolli and Dalla Fontana, 2009). We generated plots of logarithmically binned contributing area versus median slope (area-slope), logarithmically binned area versus median curvature (area-curvature), and linearly binned curvature versus median slope (curvature-slope) – all separated upstream and downstream of the knickpoint. For area-slope plots the gradient at the graphical rollover in binned area is recorded along with this area bin. We also attempted rollover identification using 2-

D kernel density estimates (Botev et al., 2010) to identify the densest concentrations of slope and area values demarcating the approximate rollover, but found similar results to the graphical approach. Following the method of Roering et al. (2007), we divide the rollover drainage area by DEM resolution to approximate the characteristic horizontal hillslope length ($L_H$), providing an additional check on DEM applicability to geomorphology. Here we use the horizontal definition of $L_H$ since the difference between horizontal and downslope $L_H$ (as measured by Grieve et al. (2016a,b,c)) should be minimal except for very

high slope angles. Area-curvature and curvature-slope plots are used to visualize the slope and area trends related to curvature, particularly around the zero curvature planar inflection point in the landscape (Roering et al., 1999).

**3.5. 2D Fourier Analysis**

In a final step, we employ a two-dimensional discrete Fourier transform (2D DFT) to quantify high-frequency noise in select datasets using 8 km by 14 km DEM clips centered on the Quebrada Honda. This common signal processing tool (Priestley,

1981) relies on the transformation of elevation matrices from the spatial to the frequency domain, providing information about the amplitude and periodicity of landscape features. Prior work using the 2D DFT on gridded topography has focused on artifact identification (Arrell et al., 2008), landscape organization and scaling (e.g., Perron et al. (2008) and references therein), the identification of landslides (Booth et al., 2009), and the length scales of biotic influence on topography (Roering et al., 2010). We follow the methods outlined in Perron et al. (2008) and take the 2D DFT of a rectangular elevation matrix, $z(x,y)$,

with $N_x \times N_y$ measurements spaced evenly by $\Delta x$ and $\Delta y$ (Priestly, 1981; Perron et al., 2008; Booth et al., 2009):

$$Z(k_x, k_y) = \sum_{m=0}^{N_x-1} \sum_{n=0}^{N_y-1} z(m\Delta x, n\Delta y) e^{-2\pi i \left( \frac{k_x m}{N_x} + \frac{k_y n}{N_y} \right)} \tag{3}$$

where $k_x$ and $k_y$ are wavenumbers and $m$ and $n$ are indices of $z$. This transformation outputs an array with the amplitudes of the frequency components, from which the power spectrum can be calculated using the DFT periodogram:

$$P_{DFT}(k_x, k_y) = \frac{1}{N_x^2 N_y^2} |Z(k_x, k_y)|^2 \tag{4}$$

The $P_{DFT}$ array is a measure of the variance of $z$ and has units of amplitude squared ($m^2$). To enhance visualization, this array is plotted against radial frequency in one-dimension as wavelength (frequency$^{-1}$) versus mean-squared amplitude. Here, the wavelength represents the spatial scale (in meters) of the amplitude fluctuations, and thus can be converted to pixel steps given the DEM resolution. A linear regression through logarithmically spaced wavelength bins is used as the background spectrum to normalize the mean-squared amplitude. Through this, we achieve a 1D plot of wavelength versus unit-less spectral power

that highlights large amplitude outliers at specific wavelengths for the 30 m SRTM-C, 12 m TanDEM-X, and 5 m AW3D5 DEMs. To test whether the DEM noise had an orientation-dependent spatial structure requiring a 2D plotting scheme, we carried out the same analysis on each elevation matrix rotated 90°, and found comparable results in the 1D plots. Further details





of this methodology, including topographic detrending and windowing functions to reduce spectral leakage, can be found in Perron et al. (2008). To quantitatively compare the 1D graphical results of the 2D DFT analysis between the 12 m TanDEM-X and 5 m AW3D5, we rely on the two sample Kolmogorov-Smirnov (KS) test (Massey, 1951) and Quantile-Quantile (QQ) plots. These statistical techniques allow the direct comparison of normalized spectral power distributions from samples with

5    different sizes (different resolutions) without the need for resampling of the data to a common resolution (e.g., 10 m), which introduces biases in the spectral analysis depending on the resampling scheme (e.g., bilinear, cubic, spline).

## 4. Results

### 4.1. Elevation Accuracy

Vertical uncertainties, measured as the mean ± SD of differences between DEM elevation and rasterized dGPS height, for all

10    DEMs are summarized in Table 2.

**Table 2. Results of pixel-by-pixel DEM vertical accuracy (DEM minus dGPS).**

| Dataset | Mean of dGPS uncertainty (m) | SD of dGPS uncertainty (m) | Number of rasterized measurements[a] | Reduction by ±30 m outlier filtering (%) |
|---|---|---|---|---|
| 30 m SRTM-C | 2.81 | 3.33 | 64,782 | 0.02 |
| 30 m AW3D30 | 1.59 | 2.81 | 63,413 | 0.03 |
| 30 m ASTER GDEM2 | -0.86 | 9.48 | 63,308 | 2.30 |
| 30 m ASTER Stack[b] | 4.56 | 6.93[c] | 15,506 | 0.12 |
| 30 m TanDEM-X | -1.29 | 2.42 | 55,791 | 0.02 |
| 12 m TanDEM-X | -1.41 | 1.97 | 108,029 | 0.02 |
| 10 m CoSSC TDX (February 7, 2011) | 1.99 | 2.02 | 28,982 | 0.03 |
| 10 m CoSSC TDX (6 November 2012) [d] | 1.32 | 3.83 | 22,182 | 0.00 |
| 10 m CoSSC TDX (25 August 2013) | 2.94 | 3.22 | 22,175 | 0.00 |
| 5 m AW3D5 | 2.40 | 1.64 | 14,306 | 0.00 |

a, After ±30 m outlier filtering
b, Generated for Pocitos Basin by weighted stacking of eight manually generated ASTER L1A DEMs
c, Compare with 11.42 m and 10.06 m SD for single L1A DEM and ASTER GDEM2 clipped to Pocitos Basin, respectively
d, CoSSC TDX DEM selected for geomorphometric analysis





Histograms of the vertical uncertainty distributions are plotted for the 30 m (Fig. 3) and higher resolution DEMs (Fig. 4). The 30 m SRTM-C and TanDEM-X both demonstrated a very low SD and smooth appearance upon visual inspections, with the TanDEM-X having the lowest SD (2.42 m) and narrowest distribution (Fig. 3A) of all 30 m DEMs. On the other hand, despite a low SD, visual inspection of numerous artifacts in the AW3D30 with no landscape representation demonstrated its lower

quality. The improvement in quality through weighted stacking of ASTER L1A stereopair DEMs versus the low quality ASTER GDEM2 is apparent in the reduction of the SD from 11.42 m for a single L1A DEM to 6.93 m for the Stack, although uncertainty distributions for all ASTER DEMs extend beyond the ±30 m outlier cutoff (Fig. 3B).

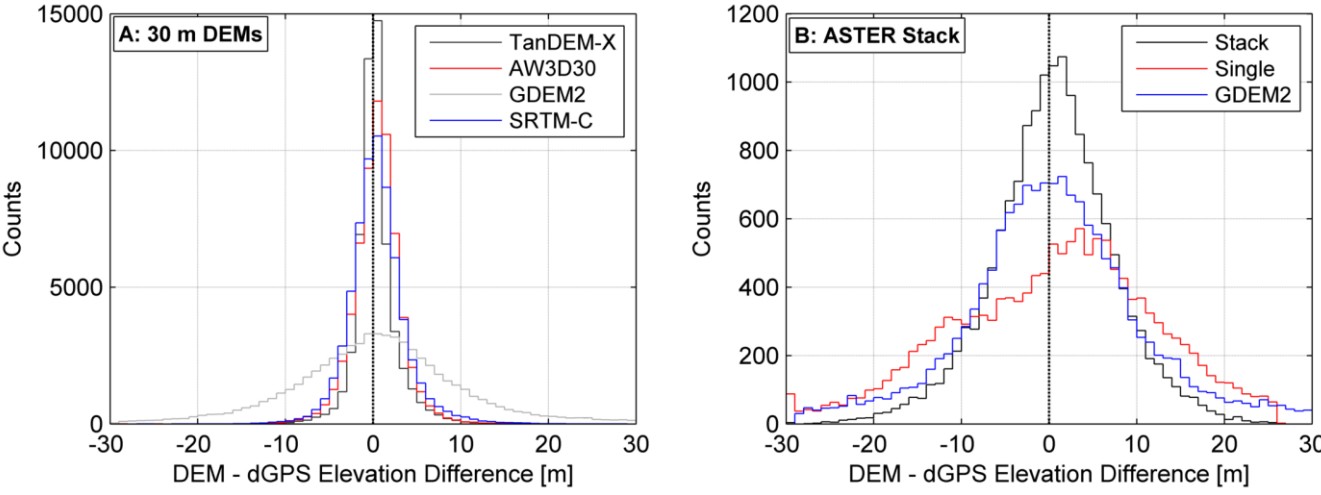

**Figure 3. Vertical uncertainties for (A) global 30 m DEMs and (B) ASTER 30 m DEMs. Plots have been normalized by mean offsets, with statistics reported in Table 2. Note the order of magnitude difference in counts, as (B) covers only the Pocitos Basin (~2,500 km², whereas (A) spans all dGPS measurements (~50,000 km²) stretching over a 4,000 m elevation range.**

For the higher resolution DEMs we note the narrow uncertainty distributions with very few ±30 m outliers (Fig. 4). The 5 m

AW3D5 has the lowest vertical SD of any DEM at 1.64 m. Importantly, the 12 m TanDEM-X DEM has a similarly low SD of 1.97 m, but covers a much larger area of dGPS measurements (~50,000 km² for the TanDEM-X versus ~580 km² for the AW3D5), as indicated by the order of magnitude difference in counts. The wider, double peaked vertical uncertainty distributions for the 6 November 2012 and 25 August 2013 CoSSC TDX DEMs are caused by their coverage over variable terrain east of the Salar de Pocitos, where accurate DEM generation is complicated by radar shadowing and layover in steeper

topography. Visual inspection of these two DEMs containing the full Quebrada Honda catchment revealed minor hillslope artifacts often coinciding with rocky outcrops and other steep and rough features, occurring in only a few areas representing a small (< 0.5 km²) portion of the catchment relative to the total area from which geomorphic metrics were derived (66 km²). Since the 2013 CoSSC TDX DEM had noticeable striping from radar processing, the 10 m CoSSC TDX DEM from 6 November 2012 was selected for further geomorphic comparison.




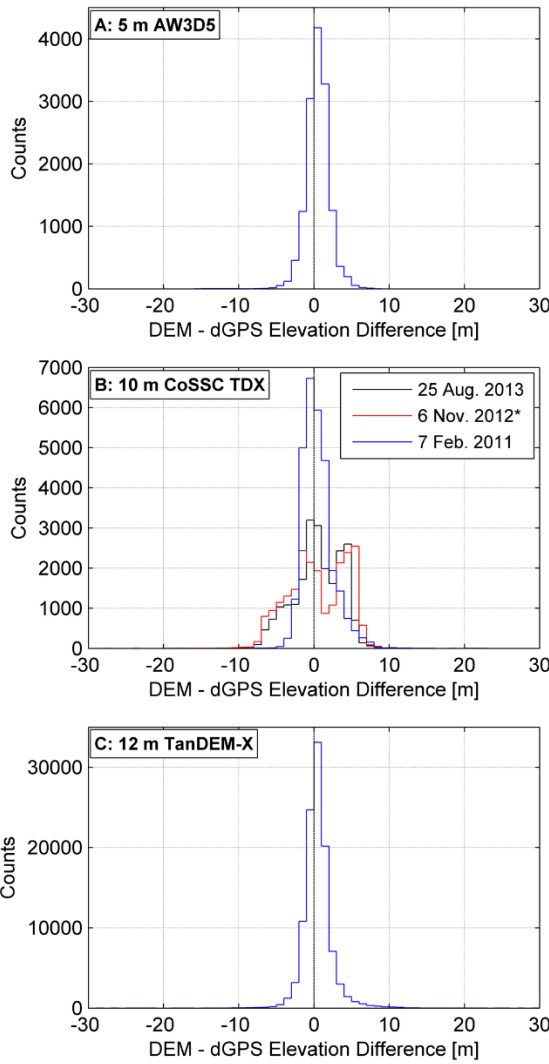

**Figure 4. Vertical uncertainties for (A) 5 m AW3D5, (B) 10 m CoSSC TDX, and (C) 12 m TanDEM-X. Plots have been normalized by mean offsets, with statistics reported in Table 2. The star (*) in (B) denotes the 2012 CoSSC TDX DEM selected for geomorphometric analysis. Note the order of magnitude increase in counts for the 12 m TanDEM-X (C), which covers nearly all dGPS measurements (~50,000 km$^2$) stretching over a 4,000 m elevation range.**

In addition to histograms, the vertical uncertainty distributions with respect to elevation, slope, and aspect of the topography are plotted for our highest quality datasets (Figs. 5-7), with additional plots in the Supplement (Figs. S4-S9). On each plot the dearth of dGPS measurements on slopes > 30° is noted, as the majority of measurements were taken from low gradient roads and flat salars. For our 30 m datasets, a narrow uncertainty range across all terrain attributes is apparent for the SRTM-C (Fig. 5), TanDEM-X (Fig. S7), and AW3D30 (Fig. S4). On the other hand, the ASTER GDEM2 (Fig. S5) had the largest error distribution across all attributes, with even low slope (0-3°) errors extending above ±10 m. Prevalent noise in the ASTER





GDEM2 is also demonstrated by the greater number of measurements (> 15,000) taken from > 10° slopes (Fig. S5C). The other 30 m DEMs provide a more realistic landscape representation with only 6,000-7,000 measurements recorded on > 10° slopes. Furthermore, the ASTER GDEM2 appears to have a slight aspect related bias with an amplitude of ~5 m repeating at approximately ENE, SE, WSW, and NNW (Fig. S5C), which is not evident in any other 30 m DEMs. We note that this aspect

bias is removed in the ASTER Stack (Fig. S6), however, this stacked 30 m DEM still has large error bars and covers only the Pocitos Basin and thus far fewer measurements than the full ASTER GDEM2. For the SRTM-C, 30 m TanDEM-X, and AW3D30, error bars grow with slope and the smallest errors are consistently found in flat topography with slopes < 10°. At low slopes (0-3°), the 30 m TanDEM-X performs exceedingly well, with 1st and 99th percentile outlier cutoffs within ±5 m (Fig. S7B). The TanDEM-X performance is further enhanced in the 12 m version (Fig. 6), which has error bars mostly within

±5 m across the range of terrain attributes (with the exception of some limited high-slope measurements). Furthermore, the 12 m TanDEM-X does not have the aspect bias noted in the 10 m CoSSC TDX DEMs (Figs. S8-S9), which have a ~5-10 m bias repeating at approximately due N, E, S, and W, likely related to satellite slant angles. Unsurprisingly from Table 2 and histogram plotting results, the 5 m AW3D5 has the smallest vertical uncertainty error bars of any DEM tested, falling almost entirely within ±5 m for all bins over all terrain attributes (Fig. 7). We emphasize that this stacked DEM was purchased for

only a small area (580 km²), but covers highly variable topography around the Nevado Queva and Quebrada Honda (Fig. 1B).





**Figure 5. 30 m SRTM-C (A) elevation, (B) slope (eight-connected neighborhood calculated), and (C) aspect (eight-connected neighborhood calculated) vertical uncertainties. Median elevation difference (black circles) with 25-75th percentile range (boxes) and 1st and 99th percentile outlier cutoff (whiskers) plotted for each bin on left axis. Number of measurements indicated (n) with measurements per bin plotted as colored circles on right axis. For aspect (C), only measurements on slopes > 10° are used, so n is reduced by an order of magnitude. Elevation differences are normalized by mean offset. We note the dearth of slope measurements > 30° (B).**







**Figure 6. 12 m TanDEM-X (A) elevation, (B) slope, and (C) aspect vertical uncertainty bias.**





**Figure 7. 5 m AW3D5 (A) elevation, (B) slope, and (C) aspect vertical uncertainty bias.**

## 4.2. Geomorphometric Analysis

5    Based on the results of elevation validation and visual inspection of the datasets, we selected the 30 m SRTM-C, 30 m and 12 m TanDEM-X, 10 m CoSSC TDX from 6 November 2012, and 5 m AW3D5 for geomorphometric analysis of the Quebrada Honda (Fig. 2). These edited (SRTM-C, TanDEM-X, and AW3D5) and single-pair radar (CoSSC TDX) DEMs, all released in the past three years, have the highest potential for future studies seeking to derive geomorphic metrics in large regions without lidar.



### 4.2.1. Channel Profile Analysis

The results of *m/n* estimation from least-squares $R^2$ maximization (Perron and Royden, 2013; Schwanghart and Scherler, 2014) and from piece-wise fitting (Mudd et al., 2014) on the Quebrada Honda trunk channel on three DEMs are summarized in Table 3. Example plots for both methods on the SRTM-C are found in the Supplement (Fig. S2-3). Not included here are the 30 m and 12 m TanDEM-X, as *m/n* results were similar to those listed. For all DEMs tested and listed in Table 1, the resulting *m/n* from least-squares fitting was 0.49-0.53 (all with $R^2 > 0.95$), representing the same range as those DEMs listed in Table 3, regardless of DEM vertical accuracy or resolution. For those DEMs tested with the piece-wise fitting method, the *m/n* range was similar, although slightly higher at 0.53-0.57. While the least-squares technique takes only a few minutes to setup and run, the computationally intensive piece-wise fitting takes hours to days, although provides a range of minimum *AICc* that denote plausible *m/n* values. Interestingly, the 5 and 10 m DEMs had slightly lower SD for *AICc*, perhaps indicating the better fitting of the channel segments compared to the 30 m DEM.

**Table 3. *m/n* values using two chi plot methods on Quebrada Honda trunk.**

| Dataset | SD of dGPS uncertainty (m) | Least-squares[a] | | Piece-wise Fitting[b] | | |
| --- | --- | --- | --- | --- | --- | --- |
| | | *m/n* | $R^2$ | *AICc* minimum value ± SD | *m/n* at *AICc* minimum | Plausible values of *m/n*[c] |
| 30 m SRTM-C | 3.33 | 0.53 | 0.97 | 27.98 ± 0.50 | 0.55 | 0.55-0.57 |
| 10 m CoSSC TDX | 3.83 | 0.49 | 0.97 | 28.89 ± 0.22 | 0.54 | -[d] |
| 5 m AW3D5 | 1.64 | 0.51 | 0.98 | 31.54 ± 0.21 | 0.54 | 0.53-0.56 |

a, Peron and Royden (2013) and Schwanghart and Scherler (2014)
b, Mudd et al. (2014)
c, With corresponding *AICc* value falling within SD range of *AICc* minimum
d, No tested values of *m/n* fell within the *AICc* SD range

### 4.2.2. Hillslope Analysis

We turn to slope (D∞), drainage area (D∞), and curvature (Laplacian) geomorphic metrics calculated for every DEM pixel upstream and downstream of the knickpoint to further assess DEM quality at a finer scale than channel analysis. Box plots showing distributions for slope and curvature are presented in Fig. 8. Here we note that the quartile (boxes) and outlier (whiskers) range of slopes are similar regardless of DEM resolution (Fig. 8A). Only the outliers grow in number and spread with fining resolution, demonstrating in particular the greater slope variability measured on the highest resolution AW3D5 DEM. In exception to this trend, more high-slope outliers are measured on the 12 m TanDEM-X compared with the 10 m





CoSSC TDX. These higher slopes are caused by the better resolution of rocky outcrops and other steep features with very high slopes from the stacked TanDEM-X DEM versus the single-CoSSC TDX DEMs. For curvature distributions (Fig. 8B), the 30 m SRTM-C and TanDEM-X, 12 m TanDEM-X, and 10 m CoSSC TDX have very narrow quartile and outlier ranges compared with the wide distribution of the 5 m AW3D5. Similar to the slope results, the higher-resolution DEMs measure far more

curvature outliers compared to the 30 m SRTM-C and TanDEM-X, whose full distributions extend only to approximately ±0.05 m$^{-1}$. Regarding differences in distributions related to the catchment knickpoint, median slopes downstream are consistently 0.1-0.2 m/m greater in magnitude than upstream (Fig. 8A). On the other hand, the downstream curvatures have a slightly narrower range compared to the upstream, especially noticeable in the smaller downstream quartile range from the AW3D5 (Fig. 8B). This is likely caused by the fact that the upstream area (48.7 km$^2$) covers more than twice as much area as

downstream (17.3 km$^2$), thus measuring more pixels and leading to a greater range in curvature measured when averaged over the entire sub-catchment area.

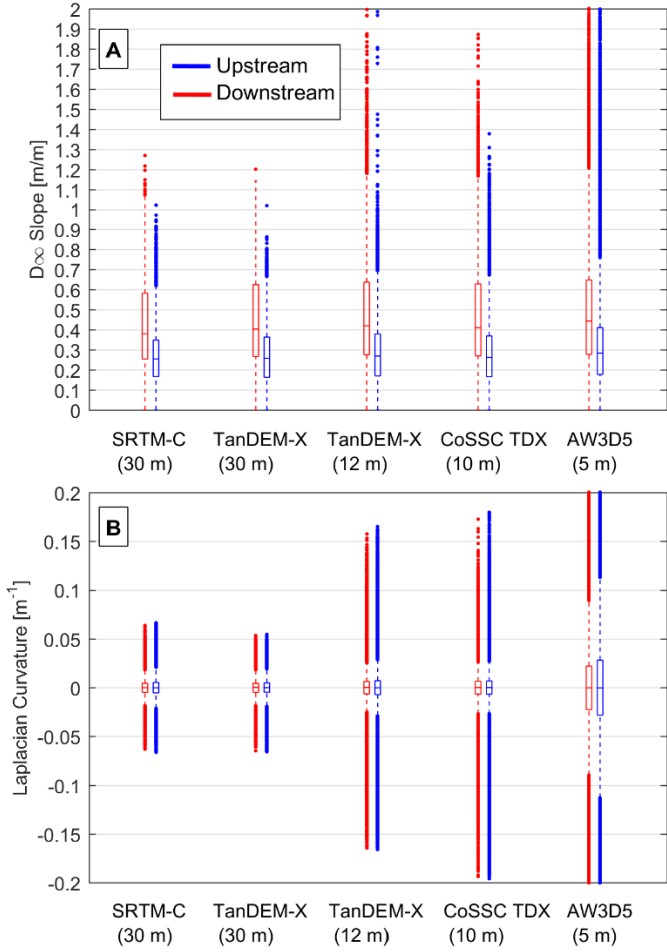

**Figure 8. Slope (A) and curvature (B) box plots separated upstream (blue) and downstream (red) for five DEMs. Center line is median, boxes are 25-75$^{th}$ percentile range, dashed whiskers extend to 1$^{st}$ and 99$^{th}$ percentiles, and all outliers are plotted as points.**
**Note that in cases where the outliers extend out of range of the plots, the points are truncated.**



Higher downstream slopes are reflected in area-slope plots with median slopes typically 0.1-0.2 m/m greater in magnitude downstream given the same contributing area (Fig. 9), regardless of DEM resolution. Moving from 30 m to 5 m, the values for median local slope (and mean slope at the rollover) change only slightly, again indicating the similar slope values measured

5   across DEMs. Differences in slope given the same drainage area are most pronounced in the $10^4$-$10^5$ m² drainage area range, with results converging somewhat as drainage area trends towards very high values, until diverging once more in the 5-12 m DEMs at drainage areas > $10^6$ m² (Fig. 9C-E). All DEMs demonstrate a broad changeover region from hillslope to fluvial processes indicated by a wide first inflection (Fig. 9). Therefore, the graphical selection of the drainage area rollover is very approximate, and upstream and downstream rollover values do not show significant variation. On the other hand, we note that

10   this rollover drainage value does decrease with fining resolution of the DEM, indicating a resolution dependent bias for this area-slope analysis, likely related to the ability of each DEM to represent the most upstream reaches of a catchment (< $10^4$ m²), which may only cover 4-5 pixels in the 30 m DEMs (900 m² per pixel) versus hundreds in the 5 m DEMs (25 m² per pixel). Nevertheless, when dividing this rollover drainage area by resolution to assess $L_H$, we find similar values across datasets. Downstream and upstream $L_H$, respectively, are 163 m and 116 m (30 m SRTM-C), 116 m and 116 m (30 m TanDEM-X), 76

15   m and 94 m (12 m TanDEM-X), 130 m and 83 m (10 m CoSSC TDX), and 119 m and 73 m (5 m AW3D5). For the 10-30 m TerraSAR-X / TanDEM-X data presented here (Fig. 9B-D), there is a noticeable "flattening" of the downstream median slopes following the initial increase, indicating that for a one to two order-of-magnitude range of drainage areas, slope values are very similar in the steeper downstream sub-catchment.



**Figure 9.** Logarithmically binned drainage area versus median slope with 25-75[th] percentile range of slopes indicated for (A) SRTM-C 30 m, (B) TanDEM-X 30 m, (C) TanDEM-X 12 m, (D) CoSSC TDX 10 m, and (E) AW3D5 5 m DEMs. Analysis separated downstream (red diamonds) and upstream (blue squares) of the knickpoint. Rollover drainage area demarcating approximate hillslope-to-valley transition after which all subsequent slope values are less is marked by vertical dashed line with slope at this bin marked by horizontal dashed line. Mean ± SD of slope (*S*) at the rollover drainage area bin (*DA*) recorded upstream and downstream. Plots are truncated to provide better visualization at very high drainage areas with low slopes.



Area-slope plots are complemented by area-curvature (Fig. 10) and curvature-slope (Fig. 11) plots to further illustrate differences related to the channel knickpoint and specific DEMs. The larger curvature variability captured in the 5 m data is again demonstrated with a larger range of curvature plotted (Fig. 10E and 11E) and larger percentile ranges about the median in each bin (Fig. 10E). Little difference in area-curvature (Fig. 10) is noticeable upstream or downstream in the 10-30 m DEMs.

5 For all DEMs, there appears to be a scaling break just below the zero-curvature planar inflection point in the concave hillslope realm, perhaps indicating the changeover between diffusive and advective processes resolvable in all DEMs (although again occurring at smaller drainage areas with fining resolution). Plots of slope binned by curvature again indicate similar slopes measured from 30 to 5 m resolution (Fig. 11). From this analysis, we note that slopes at high convex (negative) and concave (positive) curvatures are similar upstream and downstream, with greater differences on more planar slopes.





**Figure 10.** Logarithmically binned drainage area versus median curvature with 25-75th percentile range of curvatures indicated for **(A) SRTM-C 30 m, (B) TanDEM-X 30 m, (C) TanDEM-X 12 m, (D) CoSSC TDX 10 m, and (E) AW3D5 5 m DEMs. Analysis separated downstream (red diamonds) and upstream (blue squares) of the knickpoint. Greater variability in curvature (larger error bars and greater y-axis range) is measured in the 5 m data. Vertical line marks approximate changeover from diffusive hillslope processes (convex curvature) to advective fluvial processes (concave curvature).**





**Figure 11. Linearly binned curvature versus median slope with 25-75th percentile range of slopes indicated for (A) SRTM-C 30 m, (B) TanDEM-X 30 m, (C) TanDEM-X 12 m, (D) CoSSC TDX 10 m, and (E) AW3D5 5 m DEMs. Analysis separated downstream (red diamonds) and upstream (blue squares) of the knickpoint. Greater variability in curvature is measured in the higher-resolution data as indicated by the growing x-axis range. Vertical line marks planar slopes dividing diffusive hillslope processes (convex curvature) to advective fluvial processes (concave curvature).**



### 4.3. 2D Fourier Analysis

The calculation of the 2D DFT on the 30 m SRTM-C, 12 m TanDEM-X, and 5 m AW3D5 provides a quantitative assessment of topographic wavelengths and noise of the DEMs. As detailed in Sect. 3.5., the normalized spectral power is plotted against wavelength for these three datasets at their native resolution (Fig. 12), with the non-normalized 1D power spectra calculated

from Eq. (4) and regression lines used for normalization shown in the Supplement (Fig. S10). In Fig. 12, a 99.9th percentile upper envelope of normalized power displays the similar trends in the distributions, with a distinct spike in normalized power at a wavelength of ~500 m and smaller secondary peaks above (at ~800 m) and below (at ~250 m) corresponding to ridge-and-valley structures in the area of the Quebrada Honda. However, what we are more interested in is the presence of a number of high peaks at approximately 10 m, 15 m, 20 m, and 40 m in the AW3D5 plot (with less pronounced expression at 25-35 m),

causing an order-of-magnitude difference in normalized spectral power scaling (Fig. 12C). These wavelengths correspond to 2-8 pixel steps in the AW3D5 data, indicating a significant high-frequency component in this optical DEM that is not found in the 12 m or 30 m radar-derived datasets.

Statistical tests help further quantify the effect of the high-power spikes in the 5 m AW3D5 data, given the overall similar shape of the spectral power distribution to the 12 m TanDEM-X. The two sample KS test, which measures the difference in

the cumulative distribution function (CDF) for each dataset, rejects the null hypothesis that both samples are taken from the same distribution at the 99 % confidence interval with a $p$-value of 0. These results suggest that the power spectra distributions of AW3D5 and TanDEM-X are significantly different despite similar elevation validation and geomorphic metric results. To better explore how these differences relate to normalized spectral power spikes shown in Fig. 12, we plot the sample quantiles against one another (Fig. 13A). Noted are the 99th, 99.9th, 99.99th, and 99.999th quantiles representing, respectively, normalized

spectral powers of 11.1, 18.4, 33.6, and 106.1 for the AW3D5 data, and 7.1, 12.5, 21.3, and 42.1 for the TanDEM-X data. Non-linear excursion from an approximately linear trend towards higher and higher normalized spectral power for the AW3D5 is caused by only a small percentage of DFT elements greater than the 99.99th quantile. For the AW3D5, the values above this quantile represent only 840 DFT elements (of a total ~$8.4 \times 10^6$), and only 210 for the TanDEM-X (of a total ~$2.1 \times 10^6$). Additionally, the inset normalized CDF plot (Fig. 13B) shows that while the median values (~0.5 normalized CDF) correspond

between the datasets, there is the greatest diversion only at very high and very low spectral powers. Figure 13B provides an additional explanation of why the KS test is unable to reject the null hypothesis. Despite following a similar trend for over 99 % of the distribution, high-frequency (low-wavelength) noise in 2-8 pixel steps causes significant differences in the 5 m AW3D5.





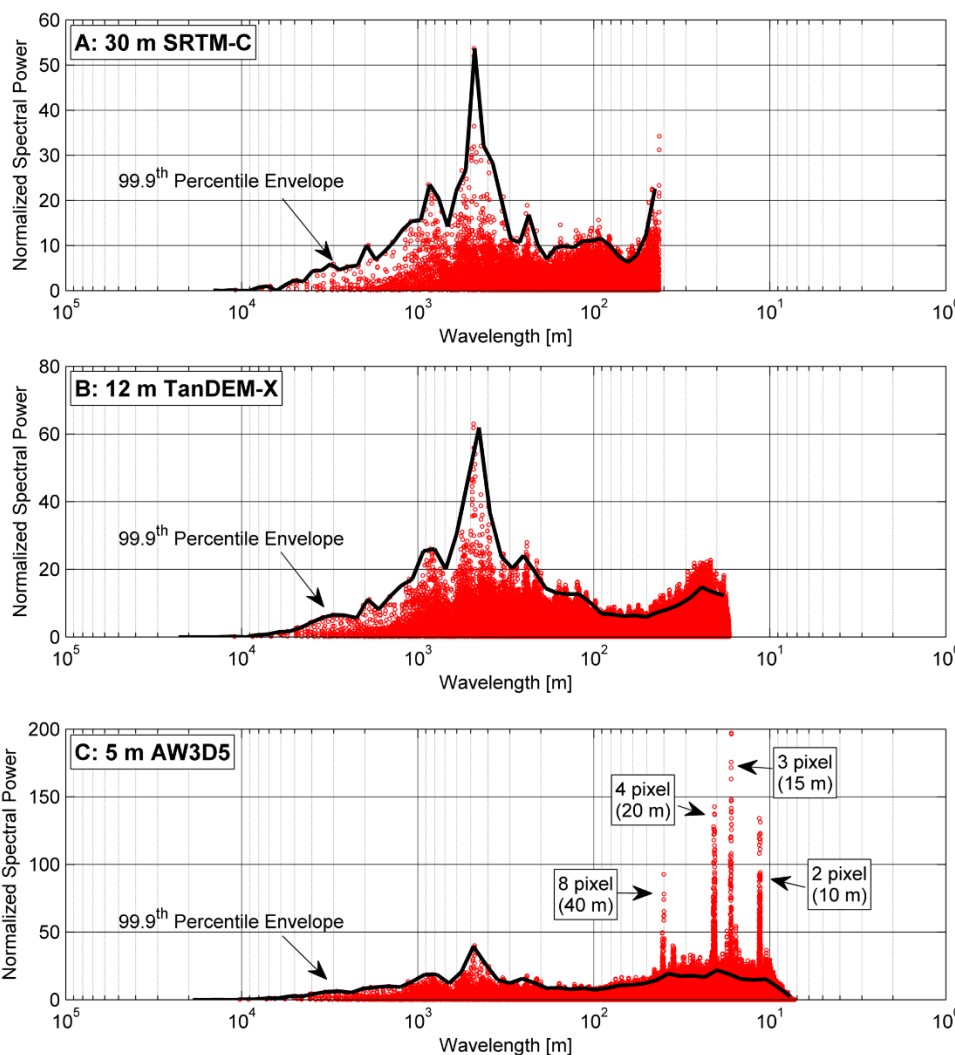

**Figure 12. 1D normalized power spectra for (A) 30 m SRTM-C, (B) 12 m TanDEM-X, and (C) 5 m AW3D5 plotted against wavelength (frequency$^{-1}$). Wavelength here is equivalent to spatial resolution in pixels. The 99.9$^{th}$ percentile envelope is calculated from 50 logarithmically spaced bins to highlight the peak in spectral power at ~500 m and the high-frequency (low-wavelength) spikes in the AW3D5 data. These spikes correspond to 2-8 pixel (10-40 m wavelength) steps in this 5 m DEM and cause an order of magnitude difference in y-scaling for plot (C).**





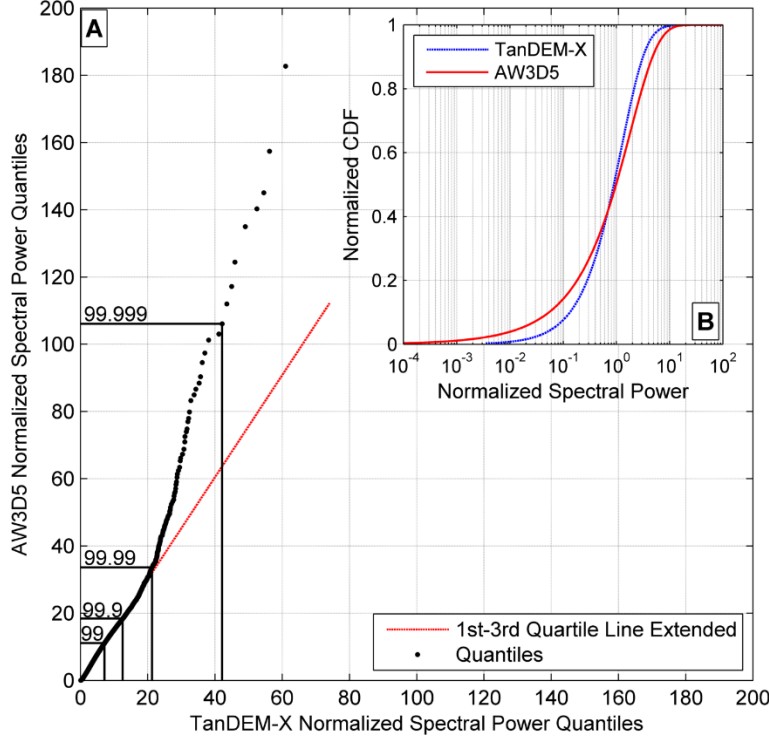

**Figure 13. Statistical analysis of 1D normalized spectral power shown in Fig. 12, with (A) QQ plot of 12 m TanDEM-X versus 5 m AW3D5 and (B) normalized CDF of normalized spectral power. Quantiles are plotted from 0.1 to 99.9998 in 0.0001 step-sizes. The 99th, 99.9th, 99.99th, and 99.999th quantiles are noted for both datasets in (A) and the linear trend connecting the 1st and 3rd quartiles is projected to display the diversion in the trend above the 99.99th quantile. We note the steeper than one-to-one trend in the data, demonstrating the higher spectral powers that dominate the AW3D5 signal.**

## 5. Discussion

### 5.1. Elevation Validation

#### 5.1.1. 30 m DEMs

The low quality of the ASTER GDEM2 is readily apparent in the wide uncertainty distribution (Fig. 3A), leading to a > 2 % outlier reduction in measurements used to assess uncertainty, but a SD remaining near 10 m (Table 2). For over 228,000 Australian National Gravity Database station heights with < 1 m vertical accuracy, Rexer and Hirt (2014) found similar results for the GDEM2 with SD ranging from 7.7 m in flat terrain to 11.29 m in mountainous regions. Other studies have reported vertical accuracies of 3-9 m for the GDEM2, but these are often determined with fewer (< 100) high accuracy control points compared with our study using over 300,000 dGPS measurements (e.g., Mukherjee et al., 2013; Athmania and Achour, 2014; Bagnardi et al., 2016). Our results indicate that in a mountainous, non-vegetated region the GDEM2 falls short of the reported vertical accuracy of 8.86 m (Tachikawa et al., 2011), even when ignoring gross outliers. In addition to the largest vertical uncertainty, the ASTER GDEM2 displays the largest uncertainties with respect to each topographic characteristic (elevation,



slope, and aspect) (Fig. S5). An increase in uncertainty is apparent with increasing slopes, indicating over prediction of elevation for the ASTER GDEM2 at higher slopes, however we also note the decrease in number of measurements at slopes > 30° (Fig. S5B). The GDEM2 experiences a clear aspect bias with an amplitude of ~5 m (Fig. S5C), which is lower than the ~50 m aspect bias reported in far-north glaciated terrain by Nuth and Kääb (2011). The ASTER Stack generated for the Pocitos

Basin shows improvement with SD reduced to 6.93 m and only 0.12 % outlier reduction (Fig. 3B), as well as elimination of the aspect bias noted in the GDEM2 (Fig. S6C). This Stack also represents an improvement over individual ASTER L1A stereopair DEMs with reported accuracies of 7-60 m (e.g., Toutin and Cheng, 2001; Hirano et al., 2003; Kääb, 2005; Nuth and Kääb, 2011). Despite this, the ASTER Stack was deemed insufficient for geomorphometry after visual inspection revealed remaining noise on hillslopes and channel elevation profiles, complicating slope and curvature measurements.

The SRTM-C, 30 m TanDEM-X, and AW3D30 have narrow vertical uncertainty distributions with SDs of 3.33 m, 2.42 m, and 2.81 m, respectively, and < 0.04 % reduction in measurements from ±30 m outlier removal (Table 2). While elevation accuracy has not been previously reported for the AW3D30 or 30 m TanDEM-X, our results indicate that these datasets exceed mission specifications of < 5 m for the AW3D30 (Tadono et al., 2014) and < 10 m for the TanDEM-X (Wessel, 2016). Most elevation accuracy reporting for the SRTM DEMs have centered on the 30 m X-band and 90 m C-band products (e.g., Rexer

and Hirt, 2014; Mukherjee et al., 2013; Kolecka and Kozak, 2014), and not the 2014 globally-released (previously only USA) 30 m C-band DEM used here. Our SRTM-C results are in close agreement with the 3.64 m accuracy found using 19 high accuracy ground measurements for a steep volcano (Bagnardi et al., 2016) and less than the 8 m accuracy versus a control DEM on another volcano (Kervyn et al., 2008). Hofton et al. (2006) report a vertical SD of 2-7 m for low vegetation regions in the USA for the SRTM-C versus high accuracy lidar data. For the 30 m SRTM-C our results exceed the 6.2 m vertical

accuracy found by Rodriguez et al. (2006) for dGPS tracks across South America.

These three high-quality 30 m DEMs exhibit no apparent biases with respect to elevation or aspect, and all show smaller ranges of uncertainties than the ASTER GDEM2. This is especially pronounced in the TanDEM-X with the narrowest uncertainty ranges plotted (Fig. S7). Vertical uncertainty at higher slopes for the SRTM-C show overestimation of elevation, in agreement with the findings of Shortridge and Messina (2011). On the other hand, the AW3D30 (Fig. S4B) and TanDEM-X (Fig. S7B)

indicate lower uncertainties (but still increasing) at these higher slopes. Previous studies suggesting SRTM-C biases related to slope and aspect (e.g., Berthier et al., 2006; Berthier et al., 2007; Van Niel et al., 2008; Shortridge and Messina, 2011) cannot be discounted by our findings, but we expect lower uncertainties with respect to slope in our non-glaciated, vegetation-free study area, where effects like radar penetration (e.g., Rignot et al., 2001; Becek, 2008; Gardelle et al., 2012) are minimal. Radar associated biases are unexplored for TanDEM-X, and are not apparent in our vegetation-free study area. These effects

are also absent from the AW3D30 as this DEM was generated through optical methods by stacking of ALOS PRISM tri-stereopairs.

Results of 30 m global DEM elevation validation indicate the high quality of height information from the SRTM-C, TanDEM-X, and AW3D30. The ASTER GDEM2 is a far noisier dataset, which complicates geomorphic analyses requiring accurate slope and curvature calculations (e.g., Kervyn et al., 2008; Fisher et al., 2013). This noise is persistent, although slightly





reduced, in the manually generated ASTER Stack. Despite its low SD, visual inspection of the AW3D30 revealed its inadequacy for assessing geomorphic metrics. Besides step-like artifacts on hillslopes, likely caused by resampling at JAXA, this dataset also had numerous holes and hillslope artifacts caused by errors in optical DEM generation. Similar to the optically generated ASTER DEMs, these errors are caused by low contrast and cloud cover that hinder stereogrammetric methods. The

30 m TanDEM-X performed best in terms of agreement with dGPS measurements and limited biases with respect to elevation, slope, and aspect. As, the SRTM-C performed comparably well in terms of elevation accuracy, both of these 30 m datasets were selected for geomorphometric analysis.

### 5.1.2. 5-12 m DEMs

Similar to the AW3D30, the vertical uncertainty for the AW3D5 exceeds the mission standard of < 5 m (Tadono et al., 2014). The low SD of 2.02, 3.83, and 3.22 m for our three CoSSC TDX DEMs are in close agreement with reported vertical accuracies of 5.74 m versus ground control points (Bagnardi et al., 2016), 3.57 m versus lidar data (Du et al., 2015), and < 2 m versus laser altimetry (Rossi et al., 2016) for interferometrically generated CoSSC TDX DEMs with resolutions of 5-12 m. Wider, bimodal uncertainty distributions for the CoSSC TDX DEMs covering the Quebrada Honda and Nevado Queva (2012 and

2013 DEMs in Fig. 4B) are likely related to radar shadowing and layover in steeper terrain. Aspect biases for these single-CoSSC radar DEMs (Fig. S8-S9) were removed in the stacked 12 m TanDEM-X relying on descending and ascending orbits, which also had a lower SD of 1.97 m, again exceeding mission standards (Wessel, 2016).

Good vertical accuracy performance is seen in the stacked 5 m AW3D5 and stacked 12 m TanDEM-X product, with both datasets having narrow vertical uncertainty ranges plotted across terrain attributes (Fig. 6-7). While interferometrically

generated single-CoSSC TDX DEMs (the same data used to generate the stacked TanDEM-X DEMs) also performed well in terms of vertical accuracy, a single stereogrammetrically generated ALOS PRISM tri-stereopair DEM (the same data used to generate the stacked AW3D DEMs) performed poorly and was not included in further analysis (see Supplement). In conjunction with the improvement seen in our ASTER Stack, these results indicate the importance of stacking multiple DEMs from the same data source to improve quality of the final product. This point is emphasized by the elimination of the aspect

bias in the stacked 12 m TanDEM-X. The higher vertical accuracy and more realistic landscape representation of the single-CoSSC TDX radar DEM versus the single ALOS PRISM tri-stereopair DEM points to the greater potential of radar to accurately represent topography (e.g., the high-quality, radar SRTM-C versus the lower quality, optical ASTER GDEM2).

Elevation accuracy for the higher resolution DEMs is similar to the high quality 30 m DEMs. The close agreement in vertical uncertainty (all < 3.5 m) between the highest quality datasets (30 m SRTM-C, 30 m and 12 m TanDEM-X, 10 m CoSSC TDX,

and 5 m AW3D5) necessitates our geomorphic metric comparisons to better understand the limitations related not only to resolution, but also to sensor. Our data shows that accurate elevation data are negligibly influenced by grid size at these resolutions (Vaze et al., 2010), making differences in DEM quality for deriving geomorphic metrics unapparent from the pixel-by-pixel dGPS comparisons and SD metric.





### 5.2. Geomorphometric Validation

#### 5.2.1. Channel Profiles

The $m/n$ values for the Quebrada Honda trunk correspond well across the datasets (30 m SRTM-C, 10 m CoSSC TDX, and 5
m AW3D5) and between the chi plot methods (Table 3). This is despite the fact that the knickpoint causes the channel to plot
non-linearly in chi space using the least-squares method (Fig. S2), whereas the piece-wise method allows exact fitting (Fig.
S3). These values (0.49-0.57) fall well within the range of reported $m/n$ values in a variety of other settings (e.g., Wobus et al.,
2006; Kirby and Whipple, 2012). Testing on the 30 m DEMs revealed similar $m/n$ values regardless of the elevation noise. For
instance, the ASTER GDEM2, which had the largest vertical uncertainty and noisiest appearance, returned $m/n = 0.53$ with $R^2$
$= 0.97$ using the least-squares method, which is identical to the SRTM-C results. The only difference for the higher resolution
datasets is a slightly lower SD (uncertainty) of minimum $AICc$ for piece-wise fitting: ~0.5 for the 30 m versus ~0.2 for the 5
and 10 m DEMs. On the other hand, the coefficient of determination ($R^2$) from least-squares fitting are nearly identical for all
three DEMs.

Differences in $m/n$ values between the datasets are likely caused by differences in channel lengths by the area threshold channel
delineation method, or by minor differences in exact channel placement downstream in the valley bottom. Nonetheless, the
$m/n$ values calculated using either chi plot method are comparable regardless of DEM resolution (or noise, as indicated by the
ASTER GDEM2 results), indicating the ability of all satellite-derived DEMs tested to resolve the valley bottom in our steep
test catchment. This result only holds for relatively simple channel shapes, like the Quebrada Honda, whereas the inclusion of
tributaries and more complex settings may warrant further testing and the preferred use of the statistically robust piece-wise
fitting method (Mudd et al., 2014). Consideration of different channel lengths and changes in $m/n$ may be an important factor
when using the ASTER GDEM2 for chi plot analysis, as this dataset has demonstrated excessive channel foreshortening over
long stretches (Fisher et al., 2013). Regardless, these results indicate that channel profile $m/n$ analysis for mapping $k_{sn}$ at $> 1$
km$^2$ scales, where minor differences related to channel head placement can be ignored (e.g., Grieve et al., 2016c), is readily
achieved on open-access 30 m DEMs.

#### 5.2.2. Hillslopes

The large increase in slope and curvature variability (outlier ranges; Fig. 8) with fining resolution can be explored in a map
view of curvature colored by a ±3-σ range for each DEM (Fig. 14). As the second derivative of elevation, curvature was
selected for map view plots to highlight variability in elevation and slope (first derivative), as elevation errors propagate to
higher derivatives. While the curvature signals of large ridges and narrow valleys are readily identified, although low in
magnitude on the 30 m DEMs (Fig. 14A-B), many more features become apparent at higher resolutions. The 12 m TanDEM-



X (Fig. 14C) and 10 m CoSSC TDX (Fig. 14D) appear similar (and have a similar ±3-σ range), although in the 10 m CoSSC TDX we note some striping becoming apparent in the second derivative of elevation, likely from interferometric processing of this single radar pair. For the 12 m TanDEM-X the hillslopes appear smooth, separated by high magnitude peaks at ridge-crests and valley bottoms. The 5 m AW3D5 (Fig. 14E) shows the greatest variability, with sharp ridges and narrow valleys becoming obscured by other high curvatures (and thus high slopes) measured across the landscape. The cause of this may be the large number of rocky outcrops visible throughout the area in the 2.5 m panchromatic, nadir ALOS PRISM optical data (Fig. 14F).

**Figure 14.** Map view of curvature from a section of the Quebrada Honda overlain on hillshade for (A) 30 m SRTM-C, (B) 30 m TanDEM-X, (C) 12 m TanDEM-X, (D) 10 m CoSSC TDX, and (E) 5 m AW3D5. Curvatures colored by a ±3-σ range to emphasize the high values, with the colorbar range noted for each DEM in the lower right. Note the striping present in the CoSSC TDX (D) and the large variability measured in the AW3D5 (E), which may be explained by the rocky outcrops apparent in the ALOS PRISM optical data for this area (F).

In Fig. 15, we explore this variability further with a map view of a hillshade image for the 5 m AW3D5 and 12 m TanDEM-X, in an area with less rocky outcrops, alongside the same area as viewed on the 2.5 m PRISM scene (the same data used to





generate the AW3D5 DEM). While ridges and valleys are similar and the debris flow gully in the center can be identified on each DEM, there is a clear difference in smoothness between the optical (AW3D5) and radar (TanDEM-X) datasets. This is exactly the noise identified in our 2D DFT analysis, which demonstrated spikes in spectral power in 2-8 pixel steps in the AW3D5 (Fig. 12C). Further, it is this high-frequency, low-wavelength noise that causes the greater number of slope outliers

and higher variability in curvature measurements, despite representing only a small (0.01 %) fraction of the power spectrum (Fig. 13). In other environments, it could be the case that this noise is caused by animal burrowing activity, tree throw, or some other high-frequency geomorphic process (e.g., Roering et al., 2010), however, it is clear in the optical data that no such processes are operating in this smooth, highly diffusive, vegetation-free environment. Rather this error is from optical DEM generation and stacking. Such high-frequency noise was also present in the optical ASTER Stack and GDEM2, as well as in

the ALOS PRISM tri-stereopair, RapidEye, and SPOT6 DEMs, all manually generated via stereogrammetry but not used for further analysis (see Supplement for details on these datasets). On the other hand, the radar-derived 12 m TanDEM-X provides a much more realistic representation of the landscape, despite a coarser resolution.

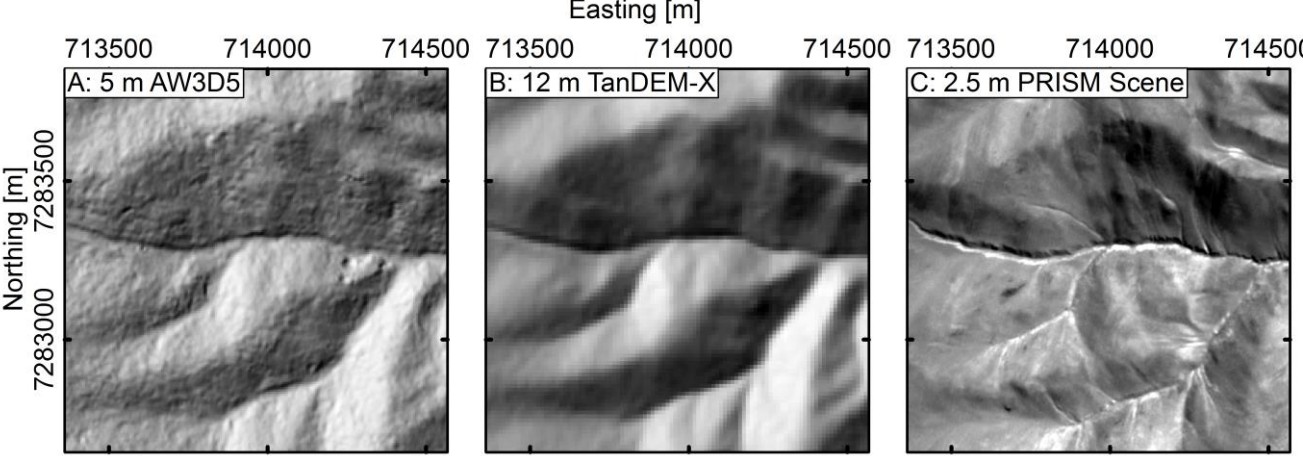

**Figure 15. Hillshade from section of the Quebrada Honda with few rocky outcrops for (A) 5 m AW3D5 and (B) 12 m TanDEM-X,**
**alongside (C) 2.5 m ALOS PRISM optical scene (nadir view). Ridges, valleys, and central debris flow gully are well represented by both DEMs, however, the high frequency noise throughout the 5 m data doesn't correspond to pit and mound topography in the optical data.**

Although curvature measurements may differ widely depending on DEM resolution (and quality), it is clear from our analysis
that slope measurements are less sensitive. All DEMs display a broad changeover between dominantly hillslope and dominantly fluvial processes, indicated by a wide area of inflection at the first rollover in area-slope plots, occurring at comparable slope values across datasets (Fig. 9). This stands in contrast to other studies using lidar DEMs, which show a very narrow changeover at DEM resolutions of 1-5 m (Tarolli and Dalla Fontana, 2009; Tseng et al., 2015). However, these studies were based in much wetter environments, where fluvial processes may act to increase this contrast. Likely, it is the case that



this hillslope-to-valley changeover is highly dependent on environmental conditions, and typically occurs over a much larger range even in the same conditions due to local complexities at small (~1 km$^2$) scales. The more pronounced difference in slope values in the $10^4$-$10^5$ m$^2$ drainage area range (Fig. 9), may indicate the increased influence of landsliding and other mass-wasting processes in the steeper downstream catchment (Tseng et al., 2015). The flattening of downstream slope values in the

TanDEM-X data in this range (Fig. 9B-D) also points to a change in geomorphic processes to more mass-wasting in the steeper downstream catchment, further indicating the high quality of this data at resolutions of 10-30 m. In area-curvature and curvature-slope plots (Figs. 10 and 11), we show a greater range of curvature measured with fining resolution, yet slope values in Fig. 11 remain comparable regardless of resolution. Further, Fig. 11 demonstrates that the 0.1-0.2 m/m magnitude difference in slope measurements upstream and downstream of the knickpoint (Fig. 9) peaks on near-planar hillslopes (zero curvature).

Hence, while valley bottoms (highly concave) and hilltops (highly convex) may have small differences in slope and curvature upstream and downstream of the kickpoint in the Quebrada Honda, the intervening hillslopes represent much of the erosional signal in the topography in this landscape.

Interestingly, the horizontal $L_H$ measured in area-slope plots differs little upstream, downstream, and between datasets. Agreement in $L_H$ between DEMs indicates that even the coarser (and noisier) DEMs are capable of measuring this key

landscape metric. With a mean of 109 ± 26 m (1-σ), $L_H$ results are within the range (though on the high side) reported in the literature using this technique (e.g., Montgomery and Foufoula-Georgiou, 1993; Roering et al., 2007; Tarolli and Dalla Fontana, 2009; DiBiase et al., 2012; Grieve et al., 2016a). The higher $L_H$ compared with other studies may be caused by the fact that the arid Central Andean Plateau, with low precipitation and little fluvial erosion (Bookhagen and Strecker, 2008; Bookhagen and Strecker, 2012), emphasizes diffusive hillslope processes. This, in turn, leads to longer hillslope measurements

compared to studies in less arid regions (e.g., Roering et al., 2007).

Given the presence of a knickpoint, we expect longer hillslopes in the gently sloped upstream catchment, whereas, our data demonstrate equally long hillslopes in the steeper downstream section. This result may be caused by the graphical selection of a rollover value in slope-area plots, which have known deficiencies in assuming uniform basin shape at these scales (Grieve et al., 2016a). However, it could be that upstream and downstream $L_H$ are in fact similar. In that case, steeper slopes downstream

given the same curvature and drainage area (Figs. 9 and 11) would be compensated by greater relief rather than longer hillslopes in order to maintain steady state of hillslopes with respect to changing baselevels caused by the migrating knickpoint. Relief measured across DEMs in a 1 km radius is consistently ~200 m greater downstream, indicating that this hypothesis is a possible explanation for the similar $L_H$.

The relative steady state of the upstream and downstream portions of the catchment would be surprising given the presence of

a migrating knickpoint, which should affect erosion rates. However, in this dominantly diffusive environment there is little advection via fluvial erosion to disturb the landscape. This hypothesis allows hillslopes to adjust to changing baselevels as the knickpoint migrates primarily during short pluvial periods on the Puna (e.g., Trauth et al., 2000; Bookhagen et al., 2001; Luna et al., in review). Although the hillslopes may adjust quickly to the changing baselevel caused by a migrating knickpoint, hilltop curvature should keep pace with erosion for much longer (Hurst et al., 2012). Despite this, it has been suggested that at




resolutions ≥ 5 m, DEMs are unable to capture the fine variability of these curvature measurements (Clubb et al., 2016). Recent work by Grieve et al. (2016c), suggests that for high accuracy DEMs, the resolution chosen for geomorphometric analysis should depend on how broad features such as ridges and valleys are, and whether that resolution can capture variations in these regions.

Attempts to examine this hilltop signal with 27 manually selected sub-basins from the 5 m AW3D5 and 12 m TanDEM-X data were complicated by transient landscape features in the Quebrada Honda. While a number of steep sub-basins can be found downstream of the knickpoint, some of these basins appear to have downstream steepened sections and upstream gently sloped sections. This is indicated by channel steepness values decreasing moving up-valley from the sub-basin outlets downstream of the knickpoint (Fig. 2A). It is possible that the trunk knickpoint signal has yet to reach the higher portions of these catchments

given the low rate of fluvial erosion on the Puna. Thus, only a limited number of sharper (more convex) hilltop measurements adjusted to the knickpoint are available to analyze sub-basin transient adjustment, which is further limited by the number of measurements available (number of pixels making up a hilltop) as resolution decreases. Furthermore, many of these hilltops are not soil-mantled (many rocky outcrops are visible in optical data) and are thus unusable for hilltop curvature measurements (Hurst et al., 2012).

Taken together, results of our study indicate that despite having low vertical uncertainties, DEMs at resolutions of 5-30 m may not all be sufficient for deriving hillslope-scale geomorphic metrics, especially curvature. This is particularly true for optical DEMs (ASTER and AW3D5), which suffer from high-frequency noise that complicates slope and curvature assessments. On the other hand, radar-derived DEMs provide a more realistic representation of the landscape, even in the case of single-CoSSC TDX DEMs. While stacking further improves DEM quality (e.g., TanDEM-X DEMs), all DEMs were capable of accurate

slope and hillslope length measurements regardless of resolution (or of noise in the case of the AW3D5). On the other hand, differences in curvature measurements (particularly outliers) were distinctly dependent on resolution. These results are in agreement with Grieve et al. (2016c), but we emphasize that apparent differences in extracted geomorphic metrics at different resolutions are not only the result of the resolution tested – as in the resampled lidar DEMs of Grieve et al. (2016c) – but also the original sensor biases and multi-step processing required to get the data in usable, gridded format. Our results lead us to

conclude that the potential of high accuracy, satellite-derived DEMs in geomorphometry and tectonic geomorphology for exploring transient responses and equilibrium adjustment of large (10-100 km$^2$) catchments remains high, however 1-5 m lidar data with realistic landscape representation may still be necessary to assess fine-scale differences in hillslope processes.

## 6. Conclusions

We were able to determine the elevation accuracy of the current generation of global 30 m digital elevation models (DEMs)

by using a 307,509 measurement differential GPS (dGPS) dataset from the high-elevation, vegetation- and cloud-free southern Central Andean Plateau (Puna de Atacama). Results indicate the high quality of the SRTM-C, TanDEM-X, and ALOS World 3D 30 m DEMs, with fewer no-data voids and fewer step-like hillslope artifacts in the SRTM-C and TanDEM-X DEMs.





Further, we have demonstrated the ASTER products' lower quality even after weighted stacking of eight meticulously generated raw L1A DEMs. We extended this analysis to the 12 m TanDEM-X DEM and demonstrated the very high quality of this dataset. In addition, we assessed the vertical accuracy of a single manually generated CoSSC TerraSAR-X / TanDEM-X 10 m DEM and the 5 m ALOS World 3D. With the exception of the ASTER data, all DEMs had vertical accuracy below 4 m. Our dGPS dataset also allowed us to explore the terrain elevation, slope, and aspect vertical biases of these DEMs. For our 30 m DEMs, we found little evidence of error biases in the SRTM-C, TanDEM-X, and ALOS World 3D, and only minor aspect biases with the ASTER GDEM2. The 12 m TanDEM-X performed exceptionally well across all terrain attributes over a large area. The 5 m ALOS World 3D performed comparably well to the TanDEM-X, although over a much smaller area. More future measurements from higher slopes ($> 30°$) would allow a fuller assessment of the previously described DEM errors on steeper topography, but are limited by dGPS measuring capabilities in the field. Both the 12 m TanDEM-X and 5 m ALOS World 3D had vertical accuracies below 2 m. The 10 m single-CoSSC TerraSAR-X / TanDEM-X DEMs displayed wide error distributions and some aspect bias, indicating the necessity of stacked scenes for accurate DEM generation (i.e., TanDEM-X). Having assessed the accuracy of the DEMs, we chose the highest quality datasets (30 m SRTM-C, 30 m and 12 m TanDEM-X, 10 m single-CoSSC TerraSAR-X / TanDEM-X, and 5 m ALOS World 3D) for a geomorphometric analysis of channel profiles and hillslopes in a large (66 km$^2$) catchment with a clear river knickpoint. We show that chi plot analysis of $m/n$ values provides comparable results regardless of DEM resolution or chi plot method when applied to the trunk channel alone in this simple setting. Regarding hillslope analyses, the 5 m ALOS World 3D (25 m$^2$ per pixel), 10 m single-CoSSC TerraSAR-X / TanDEM-X (100 m$^2$), 12 m TanDEM-X (144 m$^2$), and 30 m SRTM-C and TanDEM-X (900 m$^2$) demonstrate similar slopes and characteristic hillslope lengths. The primary difference in geomorphic analysis came in the measurement of curvature, which is shown to be highly resolution (and noise) dependent. Despite low vertical uncertainty, the ALOS World 3D data was found to be excessively noisy in 2-8 pixel steps via a Fourier frequency analysis. This high-frequency, low-wavelength noise is caused by difficulties in optical DEM generation via stereogrammetry, whereas, the interferometric radar-derived TanDEM-X DEM had much smoother, more realistic landscape representation with no spurious high-frequency noise.

We demonstrate that the newer generation of 5-12 m DEM products can be useful in assessing hillslope parameters at larger scales and lower costs than lidar, but may still be insufficient for fine-scale analysis of hilltop curvature. Caution should be taken by geomorphologists when using noisy optically generated DEMs (e.g., ALOS World 3D), when even single non-stacked radar DEMs (e.g., single-CoSSC TerraSAR-X / TanDEM-X) may provide more realistic landscape representation. DEMs acquired by remote sensing technology onboard satellites are reaching better and higher potential for geomorphic analyses including landslide detection and identification of smaller scale features. Despite this improvement, the variability of mountainous landscapes and identification of transient responses in the realm of tectonic geomorphology benefit not only from vertical accuracy, but also DEM resolution and the ability to derive higher-order derivatives.



## 7. Code Availability

Code for discrete Fourier transform analysis of DEM noise is available at https://github.com/bpurinton/DEM_fourier_noise and via the Universität Potsdam Remote Sensing account at https://github.com/UP-RS-ESP.

## 8. Data Availability

5   SRTM and ASTER data may be downloaded free of charge with links in Table 1. The same is true of the resampled ALOS World 3D 30 m DEM. The original 5 m ALOS World 3D DEM, ALOS PRISM optical tri-stereopair, and SPOT6 optical scenes were commercially purchased for this study and cannot be redistributed. Similarly, the TerraSAR-X / TanDEM-X and RapidEye data were received via multiple research proposals, which do not allow data sharing. Finally, the dGPS dataset used in this study is part of ongoing research and therefore unavailable to the public.

## 9. Supplement Link

purinton_ESD_supplement.pdf

## 10. Author Contribution

B. Bookhagen provided field equipment and training as well as financial support and supervision of analysis. B. Purinton carried out analyses, produced figures, and prepared the manuscript under B. Bookhagen's guidance.

## 11. Competing Interests

The authors declare that they have no conflict of interest.

## 12. Acknowledgements

CoSSC TerraSAR-X / TanDEM-X radar data was received through proposal ID XTI_GEOL6727 granted to B. B., and TanDEM-X DEMs were received through proposal ID DEM_CALVAL1028 granted to B. P., both from the DLR. RapidEye
20   scenes were received through RapidEye Science Archive (RESA) proposal ID 00195 granted to B. P. Ricardo Alonso, Manfred Strecker, Lisa Luna, and Patrick Lanouette are thanked for help with field work in March 2015-2016. We particularly thank Kanayim Teshebaeva for help in generating the interferometric TerraSAR-X / TanDEM-X DEMs and Wolfgang Schwanghart for development of the terrain analysis toolbox and productive conversations throughout preparation of the manuscript.





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
