# Peer review of "Validation of digital elevation models (DEMs) and comparison of geomorphic metrics on the southern Central Andean Plateau"

_Earth Surface Dynamics, 2017_

## Referee Comment (RC1) · S.W.D. Grieve (Referee) · 7 Feb 2017

I would firstly like to thank the authors for this excellently written contribution, it was very interesting to read and provides a much needed analysis of the utility of modern satellite derived elevation products for geomorphic analysis. The figures and tables are clear and well constructed and the additional information provided in the supplement greatly clarifies much of the technical work undertaken in this analysis. This manuscript compiles a collection of optical and radar derived DEM products from both open access and commercial sources and evaluates their accuracy against a

large collection of dGPS points. From this accuracy assessment a subset of the DEM products are selected upon which to perform a series of geomorphic measurements to test the applicability of these products for modern geomorphic analysis. The geomorphic evaluation of the DEMs applies both hillslope and channel metrics and demonstrates the potential and limitations of using spaceborne platforms to acquire elevation data for geomorphic analysis. An additional piece of analysis which is very valuable is the 2D Fourier frequency analysis, used to identify high frequency noise in the optical datasets, which has previously been identified as having a geomorphic origin, but in this instance appears to be solely derived from instrument error. This paper has the potential to become a valuable contribution to the discipline not only from the wide relevancy of the analysis and results to the surface processes community, but also due to its scientific rigor and clarity.

**General comments**

Overall I have no major concerns surrounding the analysis or the presentation of the analysis and consider the manuscript to be close to being ready for publication. However, I offer the following comments:

I found the abstract very dense, in particular because of the use of a large number of acronyms. While I recognize that the results for individual data products need to be specified, it may help the reader get to grips with the aims of the paper, if in addition to grouping the datasets in the 4th line of the abstract by resolution, they were also grouped into radar and optical sources, as occurs later in the manuscript.

In the discussion surrounding the measurement of hillslope length, the challenges of

interpreting hillslope length from slope area plots are highlighted clearly and although I prefer the use of a flow path method to discern hillslope length I appreciate the utility of the slope area method in this study and believe it provides some very interesting results. However, one additional issue I would like to see highlighted is the assumption that grid resolution is equivalent to the unit contour width and can be used to convert a drainage area into a characteristic hillslope length. In some cases this may be appropriate, but these two parameters are distinct and to my knowledge no work has been done to attempt to correlate these parameters. I do not expect your analysis to change, as varying a constant in the calculation of hillslope length will not change the trends of your results, however, adding a sentence to highlight this assumption within the methodology on Page 12 would enhance the clarity of this section.

Throughout the analysis, curvature and slope is calculated using a 9 cell window, which suggests that as the grid resolution is varied between data products, these derivatives of elevation will be calculated across differing length scales, potentially capturing the signals of processes operating at distinct spatial scales. I would be interested to see a small discussion of this difference between this paper's approach and other approaches to measuring curvature and slope from kernels of a variable radius.

In the line by line comments below I have identified some confusion between the terminology of grid resolution increasing or grid size decreasing. Please check the manuscript for any other instances of this.

A lot of reference is made to supplemental figures S4 to S9, I appreciate that these

are large figures, but it may be more valuable to present this information in the main manuscript to ensure readers who don't always read supplements will still see the interesting results from these datasets. However, I will leave this up to the authors and the AE to decide whether this will result in too many figures in the main manuscript.

**Line by line comments**

In addition to the issues mentioned above, I have some more general minor line by line comments:

Page 2, Line 23 - There have been developments in the production of adaptive resolution DEMs (e.g. Liu et al., 2014), it would make this section more complete to direct an interested reader to some of these papers.

Page 2, Line 31 - Grid resolution is increasing, grid size is decreasing.

Page 3, Line 26 - With this list of lidar applications it would be better if the references were placed alongside their examples, rather than a long list of uses followed by a long list of references.

Page 32, Line 7 - The global compilation of m/n values and other properties by Harel et al. (2016) would be a good reference to add in here to place these results in their full context.

Page 35, Line 27 - Is there a reason for the selection of a 1 km radius for the estimation of relief?

Page 39, Line 4 - This is the only paper title in the reference list which is in

block capitals.

Page 39, Line 8 - Check the formatting of the author's name for this paper.

**Figures and Tables**

Figure 10 - As the points obscure each other, would it be possible to introduce some transparency to the diamonds to more clearly show both datasets?

**Supplement**

The supplement is an excellent addition to the main text and provides detailed information on the data and the methodologies employed. Everything is very clear, aside from the description of the use of "standard GIS tools", it would be helpful to indicate which program you used to help future authors reproduce your work.

I have also gone through the provided Matlab code and although I have not run it as I do not have access to the right licenses, from a close reading of the code it appears to implement the analysis which is described in the paper. I would also like to thank the authors for sharing their code.

– Stuart Grieve

**References**

M-A Harel, SM Mudd, and M Attal. Global analysis of the stream power law parameters based on worldwide 10 be denudation rates. Geomorphology, 268:184–196, 2016.

[Figure]

Zhaoqin Liu, Man Peng, and Kaichang Di. A continuative variable resolution digital elevation model for ground-based photogrammetry. Computers Geosciences, 62:71–79, 2014.

---

## Referee Comment (RC2) · Anonymous Referee #2 · 10 Feb 2017

This paper presents an interesting contribution to the journal. The authors provide a detailed overview of DEMs from different sources (commercial and open), at different resolution, and they provide an evaluation of the DEM quality compared to a large dataset of dGPS point, as well as a further analysis on the possible quality of derived topographic parameters.

I have overall some minor comments that should help to improve this work before publication. Other than these minor comments, I found the paper was very well written and interesting, providing useful guidelines for geomorphometry researchers in the

future, when dealing with this type of data.

Abstract:

I think the abstract is quite complex and its complexity prevents the reader from really gathering the purpose of the paper. I suggest the authors clarify better the aim of the study and organise the results presented by, for example, DEM resolution, rather than specifying the analysis for each DEM source. i believe the authors could skip the exact measurements of the errors in the abstract, in favor of a more general overview of their results. This should help improving the readibility of the abstract.

Introduction:

I suggest some further scientific literature to consider, that is in my opinion important to provide a complete framework for this study, and might also help in improving the discussion when comparing this work to others. Recent challenges in geomorphometry have been shown in (Sofia et al., 2016). As well, aside from transient landscapes (Andreani and Gloaguen, 2016) and channel network analysis, new challenges in geomorphometry includes modelling anthropogenic landscapes (Tarolli, 2014; Passalacqua et al., 2015; Tarolli and Sofia, 2016). Concerning DEM errors, numerous studies provide interesting analysis, both on DEMs themselves and on the derived topographic parameters such as slope, curvature or other attributes (Albani and Klinkenberg, 2003; Albani et al., 2004; Raaflaub and Collins, 2006; Temme et al., 2006; Xuejun and Lu, 2008; Heritage et al., 2009; Fisher et al., 2013; Sofia et al., 2013)

Methods:

I wonder why the authors only considered a simple analysis based on the SD of residual, and do not consider a complete analysis of errors such as that presented for example in (Höhle and Höhle, 2009). I am also curious to see the differences in errors before filtering the outliers. A further commenting on what DEM presented the highest changes in accuracy after filtering should be done, to provide the reader with an idea

of the general quality of the datasets as well.

Results:

In some instances, I found a bit of confusion between grid resolution increasing/ grid size decreasing, please double check on this.

Albani M, Klinkenberg B. 2003. A Spatial Filter for the Removal of Striping Artifacts in Digital Elevation Models. Photogrammetric Engineering Remote Sensing 69: 755–765 Albani M, Klinkenberg B, Andison DW, Kimmins JP. 2004. The choice of window size in approximating topographic surfaces from Digital Elevation Models. International Journal of Geographical Information Science 18 (6): 577–593 DOI: 10.1080/13658810410001701987 Andreani L, Gloaguen R. 2016. Geomorphic analysis of transient landscapes in the Sierra Madre de Chiapas and Maya Mountains (northern Central America): implications for the North American–Caribbean–Cocos plate boundary. Earth Surface Dynamics 4 (1): 71–102 DOI: 10.5194/esurf-4-71-2016 Fisher GB, Bookhagen B, Amos CB. 2013. Channel planform geometry and slopes from freely available high-spatial resolution imagery and DEM fusion: Implications for channel width scalings, erosion proxies, and fluvial signatures in tectonically active landscapes. Geomorphology 194: 46–56 DOI: 10.1016/j.geomorph.2013.04.011 Heritage GL, Milan DJ, Large ARG, Fuller IC. 2009. Influence of survey strategy and interpolation model on DEM quality. Geomorphology 112 (3–4): 334–344 DOI: 10.1016/j.geomorph.2009.06.024 Höhle J, Höhle M. 2009. Accuracy assessment of digital elevation models by means of robust statistical methods. ISPRS Journal of Photogrammetry and Remote Sensing 64 (4): 398–406 DOI: 10.1016/j.isprsjprs.2009.02.003 Passalacqua P, Belmont P, Staley DM, Simley JD, Arrowsmith JR, Bode CA, Crosby C, DeLong SB, Glenn NF, Kelly SA, et al. 2015. Analyzing high resolution topography for advancing the understanding of mass and energy transfer through landscapes: A review. Earth-Science Reviews 148: 174–193 DOI: 10.1016/j.earscirev.2015.05.012 Raaflaub LD, Collins MJ. 2006. The effect of error in gridded digital elevation models on the estimation of
topographic parameters. Environmental Modelling Software 21 (5): 710–732 DOI: http://dx.doi.org/10.1016/j.envsoft.2005.02.003 Sofia G, Hillier JK, Conway SJ. 2016. Frontiers in Geomorphometry and Earth Surface Dynamics: Possibilities, Limitations and Perspectives. Earth Surface Dynamics 4: 721–725 DOI: 10.5194/esurf-4-721-2016 Sofia G, Pirotti F, Tarolli P. 2013. Variations in multiscale curvature distribution and signatures of LiDAR DTM errors. Earth Surface Processes and Landforms 38 (10): 1116–1134 DOI: 10.1002/esp.3363 Tarolli P. 2014. High-resolution topography for understanding Earth surface processes: Opportunities and challenges. Geomorphology 216: 295–312 Tarolli P, Sofia G. 2016. Human topographic signatures and derived geomorphic processes across landscapes. Geomorphology 255: 140–161 DOI: 10.1016/j.geomorph.2015.12.007 Temme AJAM, Schoorl JM, Veldkamp A. 2006. Algorithm for dealing with depressions in dynamic landscape evolution models. Computers Geosciences 32 (4): 452–461 DOI: http://dx.doi.org/10.1016/j.cageo.2005.08.001 Xuejun L, Lu B. 2008. Accuracy Assessment of DEM Slope Algorithms Related to Spatial Autocorrelation of DEM Errors. In Advances in Digital Terrain Analysis SE - 16, Zhou Q, , Lees B, , Tang G (eds).Springer Berlin Heidelberg; 307–322. DOI: 10.1007/978-3-540-77800-4$_{16}$
* * *

---

## Author Comment (AC1) · 24 Feb 2017

Response to Reviewer Stuart Grieve for manuscript submission to Earth Surface Dynamics,

**Validation of digital elevation models (DEMs) and comparison of geomorphic metrics on the southern Central Andean Plateau (esurf-2017-4)**

We appreciate this review and we are thankful for the insights and close reading of the manuscript. Highlighted below in **bold are the reviewer comments**. Below is our response, which addresses all changes in the final manuscript submission following completion of the interactive review period.

**General comments**

**I found the abstract very dense, in particular because of the use of a large number of acronyms. While I recognize that the results for individual data products need to be specified, it may help the reader get to grips with the aims of the paper, if in addition to grouping the datasets in the 4th line of the abstract by resolution, they were also grouped into radar and optical sources, as occurs later in the manuscript.**

The density of the abstract reflects the density of the study, but we appreciate the comments with understanding the scope given all the information (and plethora of acronyms). The second reviewer suggested simplification of the abstract, with less reference to specifics of the results. We've chosen to meet the requests somewhere in between and have removed several abbreviations from the abstract. Below we show what we hope is a simplified version of the abstract, which perhaps adds the necessary clarity:

"

In this study, we validate and compare elevation accuracy and geomorphic metrics of satellite-derived digital elevation models (DEMs) on the southern Central Andean Plateau. The plateau has an average elevation of 3.7 km, and is characterized by diverse topography and relief, lack of vegetation, and clear skies that create ideal conditions for remote sensing. At 30 m resolution, the

SRTM-C, ASTER GDEM2, stacked ASTER L1A stereopair DEM, ALOS World 3D, and TanDEM-X have been analyzed. The higher resolution datasets include 12 m TanDEM-X, 10 m single-CoSSC TerraSAR-X / TanDEM-X DEMs, and 5 m ALOS World 3D. These DEMs represent the state-of-the-art for optical (ASTER and ALOS) and radar (SRTM-C and TanDEM-X) spaceborne sensors.

We assessed vertical accuracy by comparing standard deviations of the DEM elevation versus 307,509 differential GPS measurements across 4,000 m of elevation. For the 30 m DEMs, the ASTER datasets had the highest vertical standard deviation at > 6.5 m, whereas the SRTM-C, ALOS World 3D, and TanDEM-X were all < 3.5 m. Higher resolution DEMs had generally lower uncertainty, with both the 12 m TanDEM-X and 5 m ALOS World 3D having < 2 m vertical standard deviation. Analysis of vertical uncertainty with respect to terrain elevation, slope, and aspect revealed the low uncertainty across these attributes for the SRTM-C (30 m), TanDEM-X (12-30 m), and ALOS World 3D (5-30 m). Single-CoSSC TerraSAR-X / TanDEM-X 10 m DEMs and the 30 m ASTER GDEM2 displayed slight aspect biases, which were removed in their stacked counterparts (TanDEM-X and ASTER Stack).

Based on low vertical standard deviations and visual inspection alongside optical satellite data, we selected the 30 m SRTM-C, 12-30 m TanDEM-X, 10 m single-CoSSC TerraSAR-X / TanDEM-X, and 5 m ALOS World 3D for geomorphic metric comparison in a 66 km$^2$ catchment with a distinct river knickpoint. Consistent *m/n* values were found using chi plot channel profile analysis, regardless of DEM type and spatial resolution. Slope, curvature, and drainage area were calculated and plotting schemes were used to assess basin-wide differences in the hillslope-to-valley transition related to the knickpoint. While slope and hillslope length measurements vary little between datasets, curvature displays higher magnitude measurements with fining resolution. This is especially true for the optical 5 m ALOS World 3D DEM, which demonstrated high-frequency noise in 2-8 pixel steps through a Fourier frequency analysis. The improvements in accurate space-radar DEMs (e.g., TanDEM-X) for geomorphometry are promising, but airborne or terrestrial data is still necessary for meter-scale analysis.
"

**In the discussion surrounding the measurement of hillslope length, the challenges of interpreting hillslope length from slope area plots are highlighted clearly and although I prefer the use of a flow path method to discern hillslope length I appreciate the utility of the slope area method in this study and believe it provides some very interesting results. However, one additional issue I would like to see highlighted is the assumption that grid resolution is equivalent to the unit contour width and can be used to convert a drainage area into a characteristic hillslope length. In some cases this may be appropriate, but these two parameters are distinct and to my knowledge no work has been done to attempt to correlate these parameters. I do not expect your analysis to change, as varying a constant in the calculation of hillslope length will not change the trends of your results, however, adding a sentence to highlight this assumption within the methodology on Page 12 would enhance the clarity of this section.**

We appreciate the flow path method for calculating hillslope length, and experimented with these calculations via the tools available on LSDTopotools (https://github.com/LSDtopotools; Grieve et al., 2016a,b). After much trial we were unable to get consistent results with the flow path method. We suspect this has to do with the relative coarseness of our data, compared with the high-resolution lidar data that these methods were developed on. Particularly in the processing step where soil-mantled, continuous hilltops are identified, we found widely varying results. Importantly, our study area does not have the extent of soil-mantled hilltops as those on which the algorithms were developed (e.g., Gabilan Mesa), and the algorithms are therefore perhaps inappropriate in our rocky, thinly soil-mantled, hyper-arid desert study area. Furthermore, we were not comfortable applying the flow path method that we had not personally developed and written the code for and are familiar with all boundary conditions. We are currently experimenting with different approaches to derive flowpaths, but none has been satisfying yet. Due to these reasons, we instead chose the simplified area-slope plotting technique. Although problematic for a number of reasons (including the use of unit contour width), we feel that this method nonetheless provided an interesting comparison of the various DEMs. Below we copy a passage to be modified on Page 12, Line 14-19 to highlight the contour width assumption:

"

Following the method of Roering et al. (2007), we divide the rollover drainage area by DEM resolution to approximate the characteristic horizontal hillslope length ($L_H$), providing an additional check on DEM applicability to geomorphology. This method relies on the assumption that DEM resolution is equivalent to unit contour width, which may be an oversimplification. Despite this caveat, the resolution, or unit contour width, serves as a constant for division and differences between values will not alter the trend of the results. We use the horizontal definition of $L_H$ since the difference between horizontal and downslope $L_H$ (as measured by Grieve et al. (2016a,b,c)) should be minimal, except for very high slope angles.

"

**Throughout the analysis, curvature and slope is calculated using a 9 cell window, which suggests that as the grid resolution is varied between data products, these derivatives of elevation will be calculated across differing length scales, potentially capturing the signals of processes operating at distinct spatial scales. I would be interested to see a small discussion of this difference between this paper's approach and other approaches to measuring curvature and slope from kernels of a variable radius.**

This is indeed an important distinction, as differences in slope, curvature, and other derivatives may not only be related to the different sensors, but also to the different resolutions utilized. By using an equally sized 9 cell window (3 by 3) across the datasets (with resolutions of 5-30 m), we are measuring different length scales. A few references (Albani et al., 2004; Sofia et al., 2013) pointing this effect out were suggested by the second reviewer, which we will include in the revised manuscript with regards to this discussion. As a matter of fact, we already did a small analysis of this effect by bilinear resampling the 5 m ALOS World 3D DEM to 10 m and 30 m, and running the analyses again (area-slope plotting, curvature and slope distribution comparison). By resampling the data to the coarser resolutions, we are essentially changing the length scale over which these factors are calculated (still in a 9 cell window). Our results show that with coarsening resolution the 5 m ALOS data still shows high-frequency noise, but overall become increasingly similar to TanDEM-X and SRTM-C. The resampled data have a similar median and range of slope and curvature values and similar slope rollover to the coarser data, though more outliers are still

measured in the resampled 5 m data. Please see the figure included below for clarification (Fig. S11 in the revised Supplement). This result indicates that the different resolutions capture different geomorphic signatures independent of sensor, which is a relevant result for the limitations of coarser data (particularly greater than ~15 m resolution). We include a new paragraph about these results in the revised manuscript discussion section (Sect. 5.2.2.) Page 35, Line 21 - Page 36, Line 6:

"

One important distinction to make with slope and curvature measurements is the window size used for calculation as differences may not only be related to the different sensors, but also to the different resolutions. By using an equally sized nine cell window ($3 \times 3$) across the datasets, we are measuring different length scales. Numerous authors (e.g., Albani et al., 2004; Sofia et al., 2013) point this effect out with regard to elevation error propagation. To test this, we bilinear resampled the 5 m AW3D5 to 10 m and 30 m, and examined the slope and curvature distributions compared to the 12 m TanDEM-X and 30 m SRTM-C. By resampling the 5 m data to the coarser resolutions, we are essentially changing the length scale over which the derivatives are calculated (still in a nine cell window). Our results show that with coarsening resolution the 5 m AW3D5 still shows high-frequency noise, particularly with respect to curvature, but overall become increasingly similar to TanDEM-X and SRTM-C (Fig. S11). This result indicates that higher resolution data captures more information even when measuring over the same length scale as coarser data. However, we demonstrate with the Fourier analysis that, in this case, the additional information is just sensor-related noise.

"

[Figure]

Figure S11. Slope (A) and curvature (B) box plots separated upstream (blue) and downstream (red) of the knickpoint. The AW3D5 DEM (native 5 m resolution) has been bilinear resampled (r) to 10 m and 30 m for comparison with TanDEM-X and SRTM-C. Slope and curvature are calculated in a 9 cell window. In (A) greater magnitude slopes in the interquartile range are still measured in the AW3D5, even after resampling, when compared to the other datasets at their respective original resolutions. Slope outliers, however, remain similar. In (B) particularly, we note that there are a greater number of outlier measurements and wider interquartile range for the AW3D5 resampled. We note that in both plots the differences between datasets becomes reduced in the 30 m resampled AW3D5 and 30 m SRTM-C, whereas the difference is more apparent in the 10 m resampled AW3D5 compared to the 12 m TanDEM-X.

**In the line by line comments below I have identified some confusion between the terminology of grid resolution increasing or grid size decreasing. Please check the manuscript for any other instances of this.**

Thanks for the note. We will be sure to alter accordingly.

**A lot of reference is made to supplemental figures S4 to S9, I appreciate that these are large figures, but it may be more valuable to present this information in the main manuscript to ensure readers who don't always read supplements will still see the interesting results from these datasets. However, I will leave this up to the authors and the AE to decide whether this will result in too many figures in the main manuscript.**

The reference to these figures is unfortunately unavoidable, however, given the length of the manuscript, we will not include S4-S9 in the main text, which would lead to 21 figures. The most important results are referenced in the text, and if the reader would like to confirm these statements, we leave it to them to go check the supplement. We would like to avoid adding too much information about the elevation accuracy into the manuscript, which may obscure the main points we wish to make about the geomorphometric potentials of the data.

**Line by line comments**

**Page 2, Line 23 - There have been developments in the production of adaptive resolution DEMs (e.g. Liu et al., 2014), it would make this section more complete to direct an interested reader to some of these papers.**

Reference added:

"

These datasets are commonly received in gridded format – rather than point cloud, triangulated irregular network (TIN), or other recently developed adaptive formats (e.g., Liu et al., 2014) – resulting in a defined measurement interval (grid resolution) that may oversimplify fine landscape variability.

"

**Page 2, Line 31 - Grid resolution is increasing, grid size is decreasing.**

Confusion over these terms has been noted throughout.

**Page 3, Line 26 - With this list of lidar applications it would be better if the references were placed alongside their examples, rather than a long list of uses followed by a long list of references.**

Changed accordingly.

**Page 32, Line 7 - The global compilation of m/n values and other properties by Harel et al. (2016) would be a good reference to add in here to place these results in their full context.**

Reference added.

**Page 35, Line 27 - Is there a reason for the selection of a 1 km radius for the estimation of relief?**

This discussion section is mostly speculation for our hillslope length results, but it definitely makes more sense for us to use the $L_H$ value we measured (109 $\pm$ 26 m) as the radius for relief measurement, as is done in the "How does grid-resolution modulate the topographic expression of geomorphic processes?" in Earth Surface Dynamics (Grieve et al., 2016c). Below is the modified passage using a more sensible relief radius:

"

Given the presence of a knickpoint, we expect longer hillslopes in the gently sloped upstream catchment, whereas our data demonstrate equally long hillslopes in the steeper downstream section. This result may be caused by the graphical selection of a rollover value in slope-area plots, which have known deficiencies in assuming uniform basin shape at these scales (Grieve et al., 2016a). However, it could be that upstream and downstream $L_H$ are in fact similar. In that case, steeper slopes downstream given the same curvature and drainage area (Figs. 9 and 11) would be compensated by greater relief rather than longer hillslopes in order to maintain steady state of hillslopes with respect to changing baselevels caused by the migrating knickpoint. Relief measured across DEMs with a 100 m radius (consistent with the measured $L_H$) is ~35 m greater downstream, indicating that this hypothesis is a possible explanation for the similar $L_H$.

"

**Page 39, Line 4 - This is the only paper title in the reference list which is in block capitals.**

Changed accordingly.

**Page 39, Line 8 - Check the formatting of the author's name for this paper.**

This is a technical report for the original ASTER release, we changed the reference to better reflect this:

"

METI/NASA/USGS: ASTER Global DEM Validation Summary Report, Tech. rep., METI/ERSDAC, NASA/LPDAAC, USGS/EROS, 2009.

"

**Figures and Tables**

**Figure 10 - As the points obscure each other, would it be possible to introduce some transparency to the diamonds to more clearly show both datasets?**

This figure will be adjusted in the revised manuscript.

**Supplement**

**The supplement is an excellent addition to the main text and provides detailed information on the data and the methodologies employed. Everything is very clear, aside from the description of the use of "standard GIS tools", it would be helpful to indicate which program you used to help future authors reproduce your work.**

Sorry for being vague there. The supplement is changed to say "using the Point to Raster tool in ArcGIS". We also tried this gridding by writing a python script using the GDAL module (no arcpy), but this was slow given the large number of points (>300,000) that needed to be averaged into a grid. Perhaps this is a task that can be revisited, but in the meantime a simple arcpy script calling on the "Point to Raster" tool works nicely and only takes a few minutes to run.

**I have also gone through the provided Matlab code and although I have not run it as I do not have access to the right licenses, from a close reading of the code it appears to implement the analysis which is described in the paper. I would also like to thank the authors for sharing their code.**

Thanks for taking a look at the code, and sorry to hear about the license issues. In the future we will move entirely in the open-source (Python) direction as we have done for other projects. Currently, the Fourier functions (and the other tools we rely on) are written quite nicely and easily accessible in Matlab.

Sincerely,

For both authors,

Ben Purinton

Universitaet Potsdam, Germany

purinton@uni-potsdam.de

**References**

Roering, J., Perron, J., and Kirchner, J.: Hillslope morphology and functional relationships between topographic relief and denudation, Earth and Planetary Science Letters, 264, 245-258, 2007.

Sofia, G., Pirotti, F., and Tarolli, P.: Variations in multiscale curvature distribution and signatures of LiDAR DTM errors, Earth Surface Processes and Landforms, 38, 1116-1134, 2013.

Grieve, S. W. D., Mudd, S. M., and Hurst, M. D.: How long is a hillslope?, Earth Surface Processes and Landforms, 41, 1039-1054, 2016a.

Grieve, S. W. D., Mudd, S. M., Hurst, M. D., and Milodowski, D. T.: A nondimensional framework for exploring the relief structure of landscapes, Earth Surface Dynamics, 4, 309-325, 2016b.

Grieve, S. W. D., Mudd, S. M., Milodowski, D. T., Clubb, F. J., and Furbish, D. J.: How does grid-resolution modulate the topographic expression of geomorphic processes?, Earth Surface Dynamics, 4, 627-653, 2016c.

Harel, M.-A., Mudd, S. M., and Attal, M.: Global analysis of the stream power law parameters based on worldwide $^{10}$Be denudation rates, Geomorphology, 268, 184-196, 2016.

Liu, Z., Peng, M. and Di, K.: A continuative variable resolution digital elevation model for ground-based photogrammetry, Computers & Geosciences, Elsevier, 79, 71-79, 2014.

---

## Author Comment (AC2) · 24 Feb 2017

Response to Reviewer Anonymous for manuscript submission to Earth Surface Dynamics,

**Validation of digital elevation models (DEMs) and comparison of geomorphic metrics on the southern Central Andean Plateau (esurf-2017-4)**

We thank the reviewer for the close consideration of our manuscript, and overall positive comments. Highlighted below in **bold are the reviewer comments**. Below each is our response, which will be changed in the final manuscript submission following completion of the interactive review period.

Abstract

**I think the abstract is quite complex and its complexity prevents the reader from really gathering the purpose of the paper. I suggest the authors clarify better the aim of the study and organise the results presented by, for example, DEM resolution, rather than specifying the analysis for each DEM source. I believe the authors could skip the exact measurements of the errors in the abstract, in favor of a more general overview of their results. This should help improving the readibility of the abstract.**

The first reviewer noted a similar issue and we have revised the abstract. We copy the revised abstract here:

"

In this study, we validate and compare elevation accuracy and geomorphic metrics of satellite-derived digital elevation models (DEMs) on the southern Central Andean Plateau. The plateau has an average elevation of 3.7 km, and is characterized by diverse topography and relief, lack of vegetation, and clear skies that create ideal conditions for remote sensing. At 30 m resolution, the SRTM-C, ASTER GDEM2, stacked ASTER L1A stereopair DEM, ALOS World 3D, and TanDEM-X have been analyzed. The higher resolution datasets include 12 m TanDEM-X, 10 m single-CoSSC TerraSAR-X / TanDEM-X DEMs, and 5 m ALOS World 3D. These DEMs

represent the state-of-the-art for optical (ASTER and ALOS) and radar (SRTM-C and TanDEM-X) spaceborne sensors.

We assessed vertical accuracy by comparing standard deviations of the DEM elevation versus 307,509 differential GPS measurements across 4,000 m of elevation. For the 30 m DEMs, the ASTER datasets had the highest vertical standard deviation at > 6.5 m, whereas the SRTM-C, ALOS World 3D, and TanDEM-X were all < 3.5 m. Higher resolution DEMs had generally lower uncertainty, with both the 12 m TanDEM-X and 5 m ALOS World 3D having < 2 m vertical standard deviation. Analysis of vertical uncertainty with respect to terrain elevation, slope, and aspect revealed the low uncertainty across these attributes for the SRTM-C (30 m), TanDEM-X (12-30 m), and ALOS World 3D (5-30 m). Single-CoSSC TerraSAR-X / TanDEM-X 10 m DEMs and the 30 m ASTER GDEM2 displayed slight aspect biases, which were removed in their stacked counterparts (TanDEM-X and ASTER Stack).

Based on low vertical standard deviations and visual inspection alongside optical satellite data, we selected the 30 m SRTM-C, 12-30 m TanDEM-X, 10 m single-CoSSC TerraSAR-X / TanDEM-X, and 5 m ALOS World 3D for geomorphic metric comparison in a 66 km$^2$ catchment with a distinct river knickpoint. Consistent *m/n* values were found using chi plot channel profile analysis, regardless of DEM type and spatial resolution. Slope, curvature, and drainage area were calculated and plotting schemes were used to assess basin-wide differences in the hillslope-to-valley transition related to the knickpoint. While slope and hillslope length measurements vary little between datasets, curvature displays higher magnitude measurements with fining resolution. This is especially true for the optical 5 m ALOS World 3D DEM, which demonstrated high-frequency noise in 2-8 pixel steps through a Fourier frequency analysis. The improvements in accurate space-radar DEMs (e.g., TanDEM-X) for geomorphometry are promising, but airborne or terrestrial data is still necessary for meter-scale analysis.

"

**Introduction**

**I suggest some further scientific literature to consider, that is in my opinion important to provide a complete framework for this study, and might also help in improving the discussion**

**when comparing this work to others. Recent challenges in geomorphometry have been shown in (Sofia et al., 2016). As well, aside from transient landscapes (Andreani and Gloaguen, 2016) and channel network analysis, new challenges in geomorphometry includes modelling anthropogenic landscapes (Tarolli, 2014; Passalacqua et al., 2015; Tarolli and Sofia, 2016). Concerning DEM errors, numerous studies provide interesting analysis, both on DEMs themselves and on the derived topographic parameters such as slope, curvature or other attributes (Albani and Klinkenberg, 2003; Albani et al., 2004; Raaflaub and Collins, 2006; Temme et al., 2006; Xuejun and Lu, 2008; Heritage et al., 2009; Fisher et al., 2013; Sofia et al., 2013)**

We thank the reviewer for pointing us to these additional references. We do note that some of these references already appear in the manuscript. The studies of Tarolli (2014) (referenced on Page 3, Line 28 and 32) and Passalacqua et al. (2015) (referenced on Page 3, Line 25 and 32) are reviews that primarily deal with lidar data, which is not the focus of this study. We reference Fisher et al. (2013) when discussing the issues surrounding the noisy ASTER GDEM2 dataset (Page 32, Line 22; Page 30, Line 34). We have selected a few (but not all) of the most relevant studies listed above to add to the revised manuscript. These references, followed by our reason for adding them, are listed below:

Albani et al. (2004) discusses the effect of different window sizes on the calculation of slope, aspect, and curvature from gridded DEMs. It is shown that errors propagate less as the window size is increased, although there is also a loss of fine-scale topographic information when increasing the window size. The first reviewer pointed out that the effects of different window sizes should be briefly discussed, and we will add this reference to that discussion.

Raaflaub and Collins (2006) is another important reference to be included in the methods section (Sect. 3.4.2.), where we discuss the calculation of slope and filtering. These authors demonstrate that errors propagate into slope calculations particularly strongly when using a steepest neighbor algorithm. In our study we calculate slope via the D-infinity algorithm (http://hydrology.usu.edu/taudem/taudem5/index.html), which is not as susceptible as nearest

neighbor calculations to these impacts, because slopes are divided between two cells. In our study, we also experimented with other routing and slope calculations, but did not find large differences to the D-infinity method.

Sofia et al. (2016) (from the present journal, Earth Surface Dynamics) provides a nice review and references for the current state of geomorphometry. We therefore include this reference early in the introduction.

Sofia et al. (2013) provides an analysis of curvature outliers in lidar datasets caused primarily by outliers. These errors in curvature measurement are reduced with increasing window size, following the results of Albani et al. (2004). Although this study is concerning high-resolution lidar data, we consider this reference relevant to our 5 m ALOS World 3D data, which shows curvature outliers that we find are related to error, and are reduced when resampling the data to coarser resolutions (approximately equivalent to increasing the window size of slope and curvature calculation). The reference will therefore be added accordingly in the discussion (Sect. 5.2.2.).

**Methods**

**I wonder why the authors only considered a simple analysis based on the SD of residual, and do not consider a complete analysis of errors such as that presented for example in (Höhle and Höhle, 2009). I am also curious to see the differences in errors before filtering the outliers. A further commenting on what DEM presented the highest changes in accuracy after filtering should be done, to provide the reader with an idea of the general quality of the datasets as well.**

Our study is not purely about DEM validation, but rather the geomorphometric potential of the state-of-the-art of satellite DEMs as well. Therefore, we chose to simplify our uncertainty analysis. We favor the use of a straightforward metric like the mean and standard deviation to provide a clear metric for comparing the elevation accuracy of the DEMs, particularly since the errors mostly follow normal distributions (cf. Fig. 3 and 4). This allows us to move into an analysis of the derivatives of elevation, without dwelling too long on the various ways that uncertainty can be

represented. However, we have compared the full elevation distribution, but note that the mean and standard deviation capture the essence. We note that the key information are the spatial correlation or consistency of the DEM data – the comparison on a pixel-by-pixel basis is not always relevant for geomorphometric studies. This is because geomorphometric studies use the spatial content of DEMs and higher-order derivatives and not absolute elevation values. Nevertheless, we reference several studies in the discussion (Sect. 5.1.) that make fuller assessments of elevation accuracy. With this in mind, we appreciate that the inclusion of the pre-filtering results may be helpful for the interested reader, and have therefore modified Table 2, shown below. We have also modified the methods section (Sect. 3.3.) to include on Page 8, Lines 21-27:
"

Differences of ±30 m were filtered out as outliers caused by bad data and processing errors, and the percentage reduction in number of measurements from this filtering is reported along with the pre-filtering mean and SD. While many other studies suggest additional statistical tests (e.g., Höhle and Höhle, 2009), our simplified method allows us to move into further analysis of the derivatives of elevation. We have compared the full error distribution, but note that the mean and standard deviation capture the essence. The key information is the spatial correlation or consistency of the DEM data because geomorphometric studies use the spatial content of DEMs and higher-order derivatives and not absolute elevation values.
"

**Table 2. Results of pixel-by-pixel DEM vertical accuracy (DEM minus dGPS). Mean and standard deviation before filtering denoted in parentheses, with value of n/a if there were no outliers filtered.**

| Dataset | Mean (m) | Standard Deviation (m) | Number of post-filtered rasterized measurements[a] | Reduction after ±30 m filtering (%) |
|---|---|---|---|---|
| 30 m SRTM-C | 2.18 (2.33) | 3.33 (13.74) | 64,782 | 0.02 |
| 30 m AW3D30 | 1.59 (1.66) | 2.81 (16.19) | 63,413 | 0.03 |
| 30 m ASTER GDEM2 | -0.15 (0.02) | 9.48 (17.65) | 63,308 | 2.30 |
| 30 m ASTER Stack[b] | 4.56 (4.58) | 6.93[c] (7.00) | 15,506 | 0.12 |
| 30 m TanDEM-X | -1.29 (-1.12) | 2.42 (14.57) | 55,791 | 0.02 |
| 12 m TanDEM-X | -1.41 (-1.31) | 1.97 (11.16) | 108,029 | 0.02 |
| 10 m CoSSC TDX (7 February 2011) | 1.99 (2.36) | 2.02 (21.26) | 28,982 | 0.03 |
| 10 m CoSSC TDX (6 November 2012) [d] | 1.32 (n/a) | 3.83 (n/a) | 22,182 | 0.00 |
| 10 m CoSSC TDX (25 August 2013) | 2.94 (n/a) | 3.22 (n/a) | 22,175 | 0.00 |
| 5 m AW3D5 | 2.40 (n/a) | 1.64 (n/a) | 14,306 | 0.00 |

a, After ±30 m outlier filtering
b, Generated for Pocitos Basin by weighted stacking of eight manually generated ASTER L1A DEMs
c, Compare with 11.42 m and 10.06 m SD for single L1A DEM and ASTER GDEM2, respectively, clipped to same area
d, CoSSC TDX DEM selected for geomorphometric analysis

**Results**

**In some instances, I found a bit of confusion between grid resolution increasing/ grid size decreasing, please double check on this.**

The first reviewer also noted this, and we have made sure to change the terminology accordingly.

Sincerely,

For both authors,

Ben Purinton

Universitaet Potsdam, Germany

purinton@uni-potsdam.de

**References**

Albani M, Klinkenberg B. 2003. A Spatial Filter for the Removal of Striping Artifacts in Digital Elevation Models. Photogrammetric Engineering Remote Sensing 69: 755–765

Albani M, Klinkenberg B, Andison DW, Kimmins JP. 2004. The choice of window size in approximating topographic surfaces from Digital Elevation Models. International Journal of Geographical Information Science 18 (6): 577–593 DOI: 10.1080/13658810410001701987

Andreani L, Gloaguen R. 2016. Geomorphic analysis of transient landscapes in the Sierra Madre de Chiapas and Maya Mountains (northern Central America): implications for the North American–Caribbean–Cocos plate boundary. Earth Surface Dynamics 4 (1): 71–102 DOI:10.5194/esurf-4-71-2016

Fisher GB, Bookhagen B, Amos CB. 2013. Channel planform geometry and slopes from freely available high-spatial resolution imagery and DEM fusion: Implications for channel width scalings, erosion proxies, and fluvial signatures in tectonically active landscapes. Geomorphology 194: 46–56 DOI:10.1016/j.geomorph.2013.04.011

Heritage GL, Milan DJ, Large ARG, Fuller IC. 2009. Influence of survey strategy and interpolation model on DEM quality. Geomorphology 112 (3–4): 334–344 DOI: 10.1016/j.geomorph.2009.06.024

Höhle J, Höhle M. 2009. Accuracy assessment of digital elevation models by means of robust statistical methods. ISPRS Journal of Photogrammetry and Remote Sensing 64 (4): 398–406 DOI: 10.1016/j.isprsjprs.2009.02.003

Passalacqua P, Belmont P, Staley DM, Simley JD, Arrowsmith JR, Bode CA, Crosby C, DeLong SB, Glenn NF, Kelly SA, et al. 2015. Analyzing high resolution topography for advancing the

understanding of mass and energy transfer through landscapes: A review. Earth-Science Reviews 148: 174–193 DOI: 10.1016/j.earscirev.2015.05.012

Raaflaub LD, Collins MJ. 2006. The effect of error in gridded digital elevation models on the estimation of topographic parameters. Environmental Modelling Software 21 (5): 710–732 DOI: http://dx.doi.org/10.1016/j.envsoft.2005.02.003

Sofia G, Hillier JK, Conway SJ. 2016. Frontiers in Geomorphometry and Earth Surface Dynamics: Possibilities, Limitations and Perspectives. Earth Surface Dynamics 4: 721–725 DOI: 10.5194/esurf-4-721-2016

Sofia G, Pirotti F, Tarolli P. 2013. Variations in multiscale curvature distribution and signatures of LiDAR DTM errors. Earth Surface Processes and Landforms 38(10): 1116–1134 DOI: 10.1002/esp.3363

Tarolli P. 2014. High-resolution topography for understanding Earth surface processes: Opportunities and challenges. Geomorphology 216: 295–312

Tarolli P, Sofia G. 2016. Human topographic signatures and derived geomorphic processes across landscapes. Geomorphology 255: 140–161 DOI:10.1016/j.geomorph.2015.12.007

Temme AJAM, Schoorl JM, Veldkamp A. 2006. Algorithm for dealing with depressions in dynamic landscape evolution models. Computers Geosciences 32 (4): 452–461 DOI: http://dx.doi.org/10.1016/j.cageo.2005.08.001

Xuejun L, Lu B. 2008. Accuracy Assessment of DEM Slope Algorithms Related to Spatial Autocorrelation of DEM Errors. In Advances in Digital Terrain Analysis SE - 16, Zhou Q, , Lees B, , Tang G (eds).Springer Berlin Heidelberg; 307–322. DOI: 10.1007/978-3-540-77800-416

---

## Editor Comment (EC1) · S. M. Mudd (Editor) · 13 Mar 2017

Two reviewers have now commented on the manuscript by Purinton and Bookhagen. It is clear from the comments that both reviewers believe this will be an important contribution, especially given the increasing availability of global datasets. The manuscript address a quite fundamental question: given we use topography to make predictions about sediment transport and hydrology, and in addition use it to assess geomorphic transport laws and possibly give us insight into both geologic hazards and uplift history, can we actually be confident that topographic metrics are reliable in the face of

current topographic data quality? This manuscript includes a detailed and systematic analysis of several datasets to answer this question and highlights the strengths and weaknesses of a number of widely available datasets.

The authors' responses to reviewer #1 and #2 clearly indicate that any suggestions will be integrated into a revised version of the manuscript. Where authors have not followed suggestions they give clear explanations why not. I appreciate the effort involved in producing the new figure S11, and compilation of the new table 2 to address questions about DEM validation. The revised abstract is clearer than the original and the results are clear.

I therefore believe this manuscript can proceed with minor revisions, and I look forward to seeing the revised version of the manuscript.

---

## Author Comment (AC3) · 19 Mar 2017

Dear Editor Dr. Mudd,

Thank you for your support through this productive review process. We are pleased with the positive comments from both reviewers, and have incorporated many changes into the revised manuscript, which we hope will improve its clarity. All documents (revised abstract, manuscript, supplement, and combined author's response) have been prepared and submitted separately.

Regards, Benjamin Purinton

---

## Author Response (AR2)

Response to Associate Editor Prof Simon Mudd for manuscript submission to Earth Surface Dynamics,

**Validation of digital elevation models (DEMs) and comparison of geomorphic metrics on the southern Central Andean Plateau**

Dear Editor Dr. Mudd,

Thanks once more for the positive review of this submission. What follows is a tracked change manuscript with the suggested final revision comments cared for. We have accommodated all suggested revisions with the exception of the changes to Fig. 5-7 (and Fig. S4-S9). Here, it was suggested that the color scheme be switched. We have experimented with both versions (original and new color scheme) and we find the original more suitable. Please see the revised (but omitted) figure below for a discussion of our reasoning. The rest of the changes can be viewed in the manuscript below, which has also been uploaded as a separate completed document. Any low-resolution figures in the submitted .pdf file will be cared for with high-resolution figures submitted at the point of finalization.

Sincerely,

For both authors,

Ben Purinton

Universitaet Potsdam, Germany

purinton@uni-potsdam.de

[Figure]

**Alternative Figure 5.** Here we have followed the editor's suggestion and switched the color scheme. However, we feel that this new figure is more difficult to look at given the overuse of color in the boxes and whiskers. On the other hand, the original figure (with the circles colored and the box plots in black) is easier to view, without the overwhelming colors. We have experimented with many different styles for this plot, but ultimately we feel that the original figures (Fig. 5-7 and Fig. S4-S9) provide a good compromise of information and simplicity.

[revised manuscript text omitted]